# Notch signaling stabilizes lengths of motile cilia in multiciliated cells in the lung

Neenu Joy[1,2], Aditya Deshpande[1,3], Sai Manoz Lingamallu[1,4], Vasam Manjveekar Prabantu[1], Chakenalli N Naveenkumar[5], Kumaraswamy Bharathkumar[5], Sukanya Bhat[1], Zabdiel Alvarado-Martinez[9], Alessandra Livraghi-Butrico[6], James S Hagood[7], Richard C Boucher[6], Daniel Lafkas[8], Kevin M Byrd[9,10], Shridhar Narayanan[5], Radha K Shandil[5], Arjun Guha[1]

**Airway multiciliated cells (MCs) maintain respiratory health by clearing mucus and trapped particles through coordinated ciliary beating. Although ciliary length progressively decreases along the proximal–distal (P-D) axis of the tracheobronchial tree, the mechanisms that maintain this gradient remain unclear. We show that canonical Notch signaling in MCs stabilizes ciliary length across airway regions. Inhibition of Notch signaling shortens tracheal cilia, lengthens distal airway cilia, abolishes the P-D gradient in ciliary length, and induces region-specific changes in gene expression. To assess how environmental factors influence this regulation, we examined germ-free mice and a model of *Mycobacterium tuberculosis* (*M. tb*) infection. Germ-free conditions did not alter ciliary architecture, whereas *M. tb* infection led to elongation of distal airway cilia accompanied by down-regulation of Notch signaling. These findings identify Notch signaling as a key homeostatic regulator that maintains ciliary length and preserves the P-D gradient in airway multiciliated cells.**

## Introduction

Multiciliated cells (MCs) bear motile cilia that beat in a concerted manner to coordinate fluid flow across epithelial tissues in the lung, brain, and reproductive tract. Motile cilia are hairlike organelles, and each cilium is a projection of the plasma membrane comprising a central microtubule scaffold with a ring of nine doublet protofilaments that surround a central pair of singlet protofilaments (Ishikawa, 2017). This cytoskeletal scaffold, or axoneme as it is called, is tethered to the plasma membrane by a cytoplasmic, centriole-derived basal body located at its base (Kumar & Reiter, 2021). The numbers of motile cilia borne by an MC vary from tens to hundreds, depending on the cell type. Dynein motors located along the ring of microtubule doublets in the ciliary shaft induce relative sliding of these protofilaments to generate a ciliary beat and drive fluid flow (Lyu et al, 2023).

The epithelial lining of the airways is a mucociliary escalator. This escalator comprises MCs and nonciliated secretory cells. Together, the cells produce airway surface fluid consisting of a cell-proximal periciliary fluid layer and a more luminal, gel-like, mucus layer. The tips of motile cilia contact the mucus layer and, with every beat, displace the mucus layer upward. The mucus and mucus-trapped particles, including inhaled pathogens, are then swallowed and cleared (Sleigh et al, 1988). An integral aspect of the escalator that is conserved across species is the pattern of MCs along the proximal–distal (P-D) axis. MCs in the trachea harbor longer cilia than the MCs in the distal airway (Toskala et al, 2005). This anatomical difference also correlates with differences in the frequencies at which cilia beat (Hayashi et al, 2005; Delmotte & Sanderson, 2006; Serra et al, 2022).

Heterogeneity in the lengths of motile cilia is a general feature of ciliated epithelia and is observed across tissues and organisms. MCs in the mucociliary epithelium of the oviduct, from fimbria to ampulla, exhibit a gradient of different lengths. Motile cilia in the fimbriae are longer (Rumery & Eddy, 1974). Similarly, analysis of ependymal cells in the third ventricle in the brain also exhibits heterogeneity in ciliary length (Lorencova et al, 2020). Organisms ranging from the unicellular paramecium to planarian epithelium

[1]Institute for Stem Cell Science and Regenerative Medicine (inStem), Bangalore, India   [2]SASTRA Deemed University, Thanjavur, India   [3]The University of Trans-Disciplinary Health Sciences and Technology (TDU), Bangalore, India   [4]Manipal Academy of Higher Education (MAHE), Manipal, India   [5]Foundation for Neglected Disease Research (FNDR), Doddaballapur, India   [6]Marsico Lung Institute/Cystic Fibrosis Research Center, University of North Carolina at Chapel Hill, Chapel Hill, NC, USA   [7]Department of Pediatrics (Pulmonology), Marsico Lung Institute, Children's Research Institute, University of North Carolina at Chapel Hill, Chapel Hill, NC, USA   [8]Immunology Discovery, Genentech Inc., South San Francisco, CA, USA   [9]Lab of Oral and Craniofacial Innovation (LOCI), Department of Innovation and Technology Research, ADA Science and Research Institute, Gaithersburg, MD, USA   [10]Department of Oral and Craniofacial Molecular Biology, Philips Institute for Oral Health Research, Virginia Commonwealth University, Richmond, VA, USA

Correspondence: arjung@instem.res.in
Daniel Lafkas's present address is Immunology, Infectious Diseases and Ophthalmology (I2O) Discovery and Translational Area, Roche Innovation Center, Basel, Switzerland

have ciliated epithelia that exhibit similar patterning (Rompolas et al, 2010; Tassin et al, 2016).

The mechanisms that pattern and maintain the mucociliary escalator are not well understood. The specification of MCs during development is orchestrated by a transcriptional hierarchy involving the proteins GEMC1, MCIDAS, FOXJ1, and RFX that act in concert with other transcription factors (Zhou et al, 2015; Terré et al, 2016; Stracker, 2019; Lewis et al, 2023). Together, these proteins induce a specialized multiciliogenic cell cycle program that coordinates production of ciliary components and assembly of cilia (Choksi et al, 2024). How this program establishes MC heterogeneity along the P-D axis is unclear. A recent study suggests Notch signaling in MCs induces a gradient of prominin-1 (PROM1, CD133) expression along the P-D axis that, in turn, regulates ciliary length (Serra et al, 2022). The Notch pathway is an evolutionarily conserved, juxtacrine signaling system that plays essential roles in the specification of cell fate (Hori et al, 2013). The pathway is activated when membrane-bound ligands (Delta-like 1,3,4, Jagged1, Jagged2) bind to membrane-bound Notch receptors (Notch1-4) and trigger cleavage of Notch intracellular domain (NICD) enabling its translocation into the nucleus. Once inside the nucleus, NICD associates with its downstream effector RBPJk, and drives the expression of its target genes (Hori et al, 2013). PROM1 is a cell surface protein with multifarious functions. Pertinently, PROM1 may inhibit ciliary growth in a dose-dependent manner (Serra et al, 2022).

Although developmental programs for MC specification are known, the processes that regulate MC homeostasis post-development are only beginning to be unraveled. Exposure of cultures of airway epithelial MCs to cigarette smoke led to a reduction in ciliary length (Leopold et al, 2009). This implies that ciliary length in differentiated MCs is not fixed and can be regulated. The decrease in ciliary length in these cultures is associated with a reduction in the expression of FOXJ1 and RFX, transcription factors that are known to regulate MC development, and in the expression of many ciliary components (Brekman et al, 2014).

We were alerted to a role of the Notch signaling pathway in MC homeostasis during our studies on the role of the pathway in the regulation of nonciliated secretory cells. Several groups, including ours, have established that Notch signaling is essential for the maintenance of nonciliated secretory club cells (CCs) in the adult lung and that down-regulation of signaling results in the transdifferentiation of the vast majority of CCs into MCs (Lafkas et al, 2015; Pardo-Saganta et al, 2015; Lingamallu et al, 2024). These studies have largely framed Notch signaling as a determinant of secretory versus MC fate, with its role in differentiated MCs remaining comparatively unexplored. Emerging transcriptomics analyses of airway epithelial populations—including datasets from Tabula Muris and independent single-cell studies—suggest that multiple Notch receptors (Notch1, Notch2 predominantly) may be expressed in MCs, raising the possibility that Notch signaling may have functions beyond the role in CCs. Consistent with this notion, we found that perturbation of Notch signaling in the adult airway epithelium resulted in striking alterations in ciliary architecture, including changes in ciliary length and disruption of the normal proximal–distal (P-D) gradient that characterizes the tracheobronchial tree. When we examined airway MCs after Notch inhibition, after transdifferentiation of CCs into MCs, we observed widespread changes in ciliary length and the complete absence of a P-D gradient. These findings led us to probe the role of Notch signaling in MCs and ciliary length and regulation of the P-D gradient.

# Results

## Antibody-mediated inhibition of Notch signaling alters ciliary length, abolishes the P-D gradient and is reversible

The lower airway epithelium of the murine respiratory tract, from trachea to terminal bronchioles (Fig 1A (i)), is composed of several epithelial cell types. MCs and CCs are the most abundant. MCs are intricately patterned along the P-D axis of the airways with cells in proximal airways possessing more numerous and longer motile cilia (Figs 1A (ii) and S1). Extensive work from multiple groups, including ours, has established that canonical Notch signaling has an indispensable role in airway homeostasis. The pathway is active in CCs and is essential for the maintenance of the CC fate (Lafkas et al, 2015; Pardo-Saganta et al, 2015; Lingamallu et al, 2024). Inhibition of Notch signaling results in the transdifferentiation of the vast majority of CCs into MCs (Lafkas et al, 2015; Lingamallu et al, 2024).

The role of Notch signaling in differentiated MCs has not been systematically examined. Notably, transcriptomic profiling of airway epithelial populations from murine lung—including datasets from Tabula Muris and independent single-cell studies (Strunz et al, 2020)—indicates that among Notch1–4, Notch2, Notch1, and possibly Notch3 are expressed in MCs, in decreasing order of abundance. Our in-house single-cell transcriptomics analyses are consistent with these observations. Immunohistochemical analyses of Notch receptor distribution in the airway epithelium have further shown that Notch2 is the predominant receptor expressed in the adult airway (Lafkas et al, 2015; Pardo-Saganta et al, 2015). Previous studies have robustly detected nuclear N2ICD in CCs, whereas nuclear Notch3 expression has been reported in subsets of CCs and MCs (Pardo-Saganta et al, 2015). Together, these data raise the possibility that Notch signaling may also be active in airway MCs.

To directly examine this possibility, we stained lung sections from adult mice (2–3 mo of age) with an antibody recognizing the intracellular domain of Notch2 (N2ICD), distinct from those used in prior studies (see the Materials and Methods section). As expected, N2ICD exhibited predominantly nuclear localization in CCs. Unexpectedly, we also observed nuclear N2ICD in MCs (Fig 1B (i, ii)), with many—if not most—MCs exhibiting nuclear signal. To determine whether this staining reflected ligand-dependent Notch activation, we treated mice with inhibitory antibodies targeting Notch1/Notch2 or their ligands Jagged1/Jagged2 (Lingamallu et al, 2024), harvested lungs 48 h later, and repeated N2ICD staining. Under these conditions, the nuclear N2ICD signal was completely lost in both CCs and MCs (Fig 1C (i, ii)), consistent with ligand-dependent activation of Notch signaling in MCs. We next assessed whether Notch activity in MCs is maintained with age. In lung

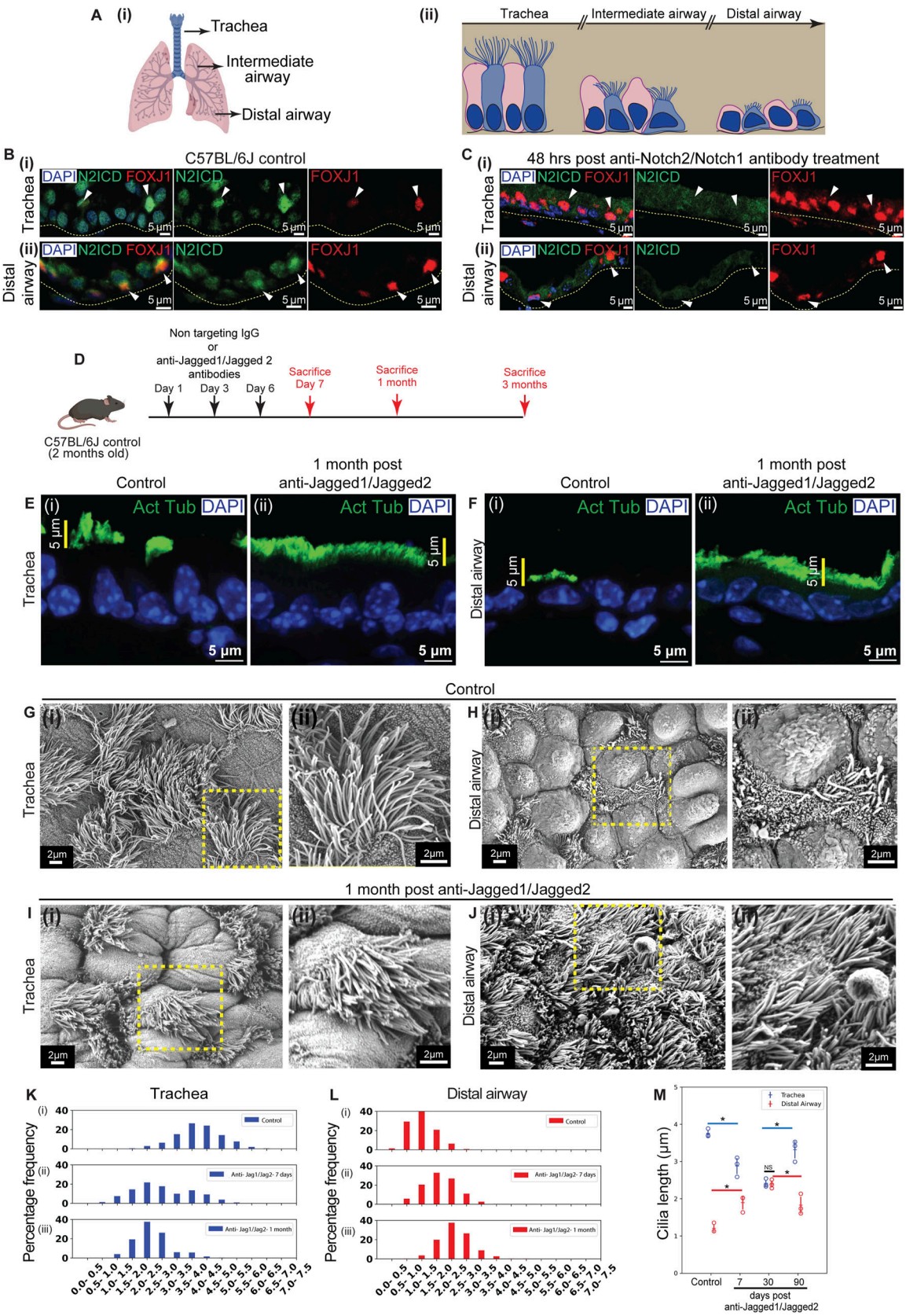

**A** **(i)** Trachea / Intermediate airway / Distal airway

**(ii)** Trachea // Intermediate airway // Distal airway

**B** **(i)** Trachea — C57BL/6J control — DAPI N2ICD FOXJ1 | N2ICD | FOXJ1

**(ii)** Distal airway — DAPI N2ICD FOXJ1 | N2ICD | FOXJ1

**C** **(i)** Trachea — 48 hrs post anti-Notch2/Notch1 antibody treatment — DAPI N2ICD FOXJ1 | N2ICD | FOXJ1

**(ii)** Distal airway — DAPI N2ICD FOXJ1 | N2ICD | FOXJ1

**D** Non targeting IgG or anti-Jagged1/Jagged 2 antibodies — Day 1 / Day 3 / Day 6 / Sacrifice Day 7 / Sacrifice 1 month / Sacrifice 3 months — C57BL/6J control (2 months old)

**E** Trachea — Control **(i)** Act Tub DAPI — 1 month post anti-Jagged1/Jagged2 **(ii)** Act Tub DAPI

**F** Distal airway — Control **(i)** Act Tub DAPI — 1 month post anti-Jagged1/Jagged2 **(ii)** Act Tub DAPI

**Control**

**G** Trachea **(i)** **(ii)**

**H** Distal airway **(i)** **(ii)**

**1 month post anti-Jagged1/Jagged2**

**I** Trachea **(i)** **(ii)**

**J** Distal airway **(i)** **(ii)**

**K** Trachea — Percentage frequency — **(i)** Control **(ii)** Anti-Jag1/Jag2- 7 days **(iii)** Anti-Jag1/Jag2- 1 month

**L** Distal airway — Percentage frequency — **(i)** Control **(ii)** Anti-Jag1/Jag2- 7 days **(iii)** Anti-Jag1/Jag2- 1 month

**M** Cilia length (µm) — Trachea / Distal Airway — Control / 7 / 30 / 90 — days post anti-Jagged1/Jagged2

sections from aged mice (1.2–1.5 yr), nuclear N2ICD was detected in virtually all MCs examined (n = 3 animals). Although we note that differences in staining patterns for Notch2, reported across studies using distinct Notch2 antibodies, remain incompletely understood and warrant further investigation, the data shown here are consistent with Notch2 activation in MCs. We also examined the distribution of other Notch receptors: Notch1 immunoreactivity was primarily membrane-localized in both CCs and MCs, whereas Notch3 staining was nuclear in subsets of both cell types, consistent with previous reports (Pardo-Saganta et al, 2015). Taken together, these data indicate that Notch signaling is active in adult airway MCs, motivating a direct investigation into its functional role in MC homeostasis and ciliary regulation.

Next, we investigated whether Notch signaling has any role in MCs. Mice were injected with either nontargeting immunoglobulin G (IgG control) or anti-Jagged1/Jagged2 antibodies (for detailed protocol, see the Materials and Methods section), and tracheae and lungs from these animals were harvested at different time points and processed for histological analysis (schematic in Fig 1D). The structure of cilia on MCs was analyzed by fluorescence and scanning electron microscopy (SEM).

Immunostaining with anti-acetylated tubulin, an antibody that detects posttranslationally modified tubulin enriched in motile cilia, clearly outlines MCs in lung sections (Figs S1 and 1E (i) and F (i)). The antibody illuminates a band of the luminal cilia, diminishing in length along the P-D axis (Figs S1D–F and 1E (i) and F (ii)). Sections from lungs treated with either nontargeting IgG or anti-Jagged1/Jagged2 antibodies were stained with anti-acetylated tubulin and examined. We detected a lawn of acetylated tubulin staining in sections from anti-Jagged–treated lungs at both 7 d and 1 mo after injection (Figs 1E (ii) and F (ii) and S2A and B). Interestingly, differences in the length of luminal staining along the P-D axis were perturbed at 7 d posttreatment (Fig S2A and B) and conspicuously absent 1 mo after antibody treatment (Fig 1E (ii) and F (ii)). These stainings suggested that the inhibition of Notch signaling had altered ciliary length in all MCs throughout the airways. Proximal cilia appeared to decrease in length, whereas distal airway cilia increased in length (compare Fig 1E (i, ii) and F (i, ii)).

To examine the effects of Notch inhibition at higher resolution, we turned to SEM. Thick sections (200 $\mu m$) of tracheal rings and lung (left lobe) were imaged. Motile cilia were observed throughout the airways, and ciliary length appeared to decrease along the P-D axis (Fig S1). Ciliary lengths in the trachea and distal airways (terminal bronchioles) were quantified from micrographs using a freehand contour-tracing approach (see the Materials and Methods section). Measurements of length demonstrated a statistically significant decrease in ciliary length from trachea through intermediate airways to terminal bronchioles (Fig S1G). Next, we compared ciliary morphology in lungs from mice that were injected with either nontargeting IgG or anti-Jagged1/Jagged2 antibodies (Fig 1G–M). Anti-Jagged treatment led to a decrease in ciliary length in the trachea and to an increase in ciliary length in the distal airway (Fig 1G–J; histograms of ciliary lengths shown in Fig 1K and L; representative images from 7 d shown in Fig S2; and dot plots of ciliary lengths in the trachea and distal airway across all time points shown in Fig 1M). No P-D gradient in ciliary length was detected at 1 mo (Fig 1M).

Previous studies have shown that the imbalance in CCs and MCs induced by anti-Jagged treatment is gradually reversed as Notch signaling recovers. Over time, variant CCs—rare CCs that resist transdifferentiation—proliferate and repopulate the airway epithelium with both CCs and MCs (Lingamallu et al, 2024). By 3 mo posttreatment, discrete patches containing CCs and MCs appear within the otherwise MC-rich epithelium (Lingamallu et al, 2024).

To trace the fate of MCs and their cilia during this long-term recovery, we employed a lineage-tracing approach. *FoxJ1*$^{\text{CreERT2/+}}$ mice, which express CreER specifically in MCs, were crossed with *Rosa*$^{\text{Tdtomato/+}}$ reporter animals. The resulting *FoxJ1*$^{\text{CreERT2/+}}$; *Rosa*$^{\text{Tdtomato/+}}$ progeny were first injected with anti-Jagged1/Jagged2 antibodies to induce MC transdifferentiation and, shortly thereafter, treated with tamoxifen to activate Tdtomato expression in both preexisting and transdifferentiated MCs (collectively referred to as "old MCs"; see experimental design in Fig S3A).

---

**Figure 1. Acute inhibition of Jagged1 and Jagged2 alters ciliary lengths and abolishes the P-D gradient in a reversible manner.**
**(A)** Diagrams of (i) lung showing trachea, intermediate airways, and distal airway, and (ii) airway epithelium showing the mucociliary escalator comprising multiciliated cells (MCs, blue) and club cells (CCs, pink). **(B, C)** Status of Notch signaling in MCs in the adult lung. **(B)** (i) Tracheal section from C57BL/6J animals showing the distribution of Notch2 intracellular domain (N2ICD, green) in MCs (stained with anti-FOXJ1, red, white arrowheads). Nuclei stained with DAPI (blue). **(B)** (ii) Lung section from C57BL/6J animals showing the distribution of N2ICD (green) in MCs (stained with anti-FOXJ1, red, white arrowheads) in the distal airway. **(C)** (i) Tracheal section from C57BL/6J animals 48 h posttreatment with anti-Notch1/Notch2 antibodies showing N2ICD (green) in MCs (red, white arrowheads). **(C)** (ii) Lung section from C57BL/6J animals 48 h posttreatment with anti-Notch1/Notch2 antibodies showing N2ICD (green) in MCs (red, white arrowheads) in the distal airway. **(D)** Regimen for inhibition of Notch signaling using nontargeting IgG (control) and anti-Jagged1/Jagged2 antibodies. **(E, F)** Ciliary staining in the trachea (E) and distal airway (F) of control (1 mo postinjection of nontargeting IgG) (i) and anti-Jagged1/Jagged2-treated (ii) lungs. **(E, F)** (i) Distribution of anti-acetylated tubulin (green) in airways from control lungs. **(E, F)** (ii) Distribution of anti-acetylated tubulin (green) in airways from anti-Jagged1/Jagged2-treated lungs. Note the pattern of ciliary staining in the trachea and distal airway in controls and the absence of this pattern 1 mo post-antibody treatment. **(G, H, I, J)** Scanning electron microscopy of sections from control (1 mo postinjection of nontargeting IgG) and anti-Jagged1/Jagged2-treated lungs. **(G, H)** Micrographs of trachea (G) (i, ii) and distal airway (H) (i, ii) from control. Magnified images of boxed regions in (i) are shown in (ii). **(I, J)** Micrographs of trachea (I) (i, ii) and distal airway (J) (i, ii) from anti-Jagged1/Jagged2-treated lungs. Magnified images of boxed regions in (i) are shown in (ii). **(K, L, M)** Quantification of ciliary length in micrographs ($\mu m$; see the Materials and Methods section). **(K, L)** Frequency distribution of ciliary lengths in the trachea and distal airways of control and anti-Jagged1/Jagged2-treated lungs. **(K)** (i–iii) Frequency distribution of ciliary lengths ($\mu m$) in the trachea (blue) in control ((i), n = 981 cilia), 7-d post–anti-Jagged1/Jagged2-treated ((ii), n = 406 cilia; see Fig S2), and 1-mo post–anti-Jagged1/Jagged2-treated ((iii), n = 365 cilia) lungs. **(L)** (i–iii) Frequency distribution of ciliary lengths ($\mu m$) in the distal airway (red) in control ((i), n = 869 cilia), 7-d post–anti-Jagged1/Jagged2-treated ((ii), n = 2,409 cilia; see Fig S2), and 1-mo post–anti-Jagged1/Jagged2-treated ((iii), n = 954 cilia) lungs. Note the alteration of ciliary lengths and absence of a P-D gradient anti-Jagged1/Jagged2-treated lungs. **(M)** Dot plots of ciliary lengths in the trachea (blue) and distal airway (red) from control lungs and anti-Jagged1/Jagged2-treated lungs 7 d, 1 , and 3 mo posttreatment (see the text for details, n = 892 cilia in trachea and n = 2,430 cilia in distal airway). Hollow circles indicate mean ciliary length in individual animals. Data in dot plot represent the mean ± SD (n = 3 animals) (NS, nonsignificant) (* denotes $P < 0.05$, t test). Note that the alteration of ciliary lengths post–anti-Jagged1/Jagged2 treatment is reversed by 3 mo. See also Figs S1, S2, and S3.

When lineage-labeled mice were analyzed 3 mo post treatment, Tdt$^+$ MCs were distributed throughout the airways, interspersed with Tdt$^-$ patches containing CCs and newly generated MCs (Fig S3B–E). These observations indicate that a large proportion of the original MCs persisted during epithelial regeneration and remained the predominant population at this time point. SEM of Tdt$^+$ MC lawns, excluding mixed CC-MC patches (Fig S3F and G), revealed that by 3 mo posttreatment, tracheal cilia had lengthened, whereas distal airway cilia had shortened relative to 1 mo (Fig 1M). Thus, restored Notch signaling coincides with reestablishment of ciliary length gradient.

Taken together, these findings led us to the following conclusions. First, MCs activate Notch signaling during homeostasis. Second, the signaling pathway regulates ciliary length throughout the airways and inhibition of the pathway abolishes the P-D gradient in ciliary length. Third, changes in ciliary length that occur upon acute Notch inhibition are reversible once signaling is restored. These findings identify Notch signaling as a regulator of ciliary architecture that maintains structural homeostasis along the P-D axis of the airways.

## Genetic ablation of *Rbpjk* in multiciliated cells (MCs) establishes a role of canonical Notch signaling in maintaining ciliary length during homeostasis

To determine whether Notch signaling in MCs directly regulates ciliary homeostasis, we adopted a genetic approach. We conditionally deleted *Rbpjk*, the gene encoding a transcription factor that mediates Notch signaling-dependent gene transcription, in MCs and analyzed the resulting phenotypes. *FoxJ1*$^{CreERT2/+}$ mice were crossed with *Rbpjk*$^{flox/flox}$ animals to generate *FoxJ1*$^{CreERT2/+}$; *Rbpjk*$^{flox/flox}$ progeny. Tamoxifen administration induced CreER activity and deletion of *Rbpjk* in MCs (hereafter referred to as *RbpjkΔMC*). Littermates that did not receive tamoxifen served as controls (experimental design in Fig 2A).

Efficient *Rbpjk* deletion was confirmed by immunostaining for RBPJκ at 10 d and 1 mo post-tamoxifen treatment (Fig S4). In controls, nuclear RBPJκ was detected in all airway epithelial cells, including MCs (Fig S4B–D), whereas staining was depleted in *RbpjkΔMC* lungs at both time points (Fig S4B, E, and F). Deletion efficiency was higher in distal airways than in the trachea and increased between 10 d and 1 mo after induction (Fig S4B).

We next assessed whether loss of *Rbpjk* affected the overall balance of MCs and club cells (CCs). Immunostaining for CC10/Scgb1a1 (CC marker) and FoxJ1 (MC marker) revealed comparable frequencies of both cell types in the trachea and distal airways of control and *RbpjkΔMC* mice at 1 mo (Fig 2B). Thus, canonical Notch signaling in MCs is dispensable for maintaining the relative proportions of airway epithelial cell types.

To examine how *Rbpjk* deletion affected MC morphology, we analyzed ciliary structure by acetylated tubulin immunostaining and SEM. At 1 mo post-tamoxifen, *RbpjkΔMC* tracheae showed shorter cilia compared with controls, whereas distal airways displayed longer cilia (Fig 2C (i, ii) and D (i, ii)). No consistent differences were observed at 10 d (Fig S5A and B). SEM imaging and quantitative analysis confirmed a decrease in tracheal ciliary

length and an increase in distal airway length over time (Fig 2E–K; histograms in Fig 2I and J; representative images in Fig S5C and D; and dot plots in Fig 2K). These findings indicate that Notch signaling in MCs stabilizes ciliary length and preserves the proximal–distal gradient.

The extent of the perturbation in *RbpjkΔMC* mice was milder than that observed after anti-Jagged1/Jagged2 treatment (compare Figs 1 and 2), likely reflecting incomplete *Rbpjk* recombination (Fig S4).

## Notch signaling maintains prominin-1 expression in adult airway multiciliated cells

To examine how Notch signaling in MCs contributes to airway homeostasis, we performed spatial transcriptomics profiling using the GeoMx Digital Spatial Profiler (DSP). This platform enables regional gene expression analysis from defined regions of interest (ROIs). Paraffin-embedded tracheal and lung sections from control (*FoxJ1*$^{CreERT2/+}$; *Rosa*$^{Tdtomato/+}$, tamoxifen-induced) and *RbpjkΔMC* (1 mo post-tamoxifen) mice were hybridized with RNA probe sets and immunostained with anti-acetylated tubulin to identify airways and MCs. ROIs were collected from trachea and distal airways (6 control trachea, 14 control distal airway, 12 *RbpjkΔMC* trachea, and 12 *RbpjkΔMC* distal airway; pipeline in Fig S6). Probe libraries from each ROI were prepared and sequenced as described (see the Materials and Methods section).

Comparison of control tracheal and distal airway ROIs revealed distinct regional transcriptomic signatures. Tdt expression was robust in control samples (Table S1), confirming inclusion of MCs in the profiled regions. Differential gene expression analysis ($|\log_2 FC| > 0.5$, $P < 0.05$) showed that tracheal and distal airway ROIs differed substantially (Fig S6B). These profiles reproduced known regional trends: *Reg3g* expression was enriched in tracheal CCs, whereas *Scgb1a1* and *Hp* were higher in distal airway CCs (Guha et al, 2014), validating the spatial resolution of the dataset.

We next compared *RbpjkΔMC* and control samples within each region. As expected, *Tdt* was absent from *RbpjkΔMC* samples (Table S1). Overall, the number of differentially expressed genes (DEGs) between tracheal and distal ROIs decreased from 5,953 in controls to 1,439 in *RbpjkΔMC* (Fig S3C), indicating that loss of Notch signaling in MCs reduces transcriptional heterogeneity along the airway.

When comparing *RbpjkΔMC* to control within each region, 505 genes were up-regulated and 1,053 down-regulated in the trachea, whereas 356 were up-regulated and 79 down-regulated in the distal airway (Fig S3D and E). Overlap between the tracheal and distal DEG sets was limited (Table S2). Many of the most significantly altered genes, such as *Hp* and *Scgb1a1/CC10*, are normally expressed in CCs rather than MCs. Cross-referencing with published single-cell RNA-seq data (Strunz et al, 2020) confirmed that several DEGs are enriched in non-MC types (Table S2). These results suggest that Notch signaling in MCs exerts a broad, non–cell-autonomous influence on airway gene expression and epithelial homeostasis.

Because Notch signaling stabilizes ciliary length (above), we next examined whether disruption of the pathway alters the

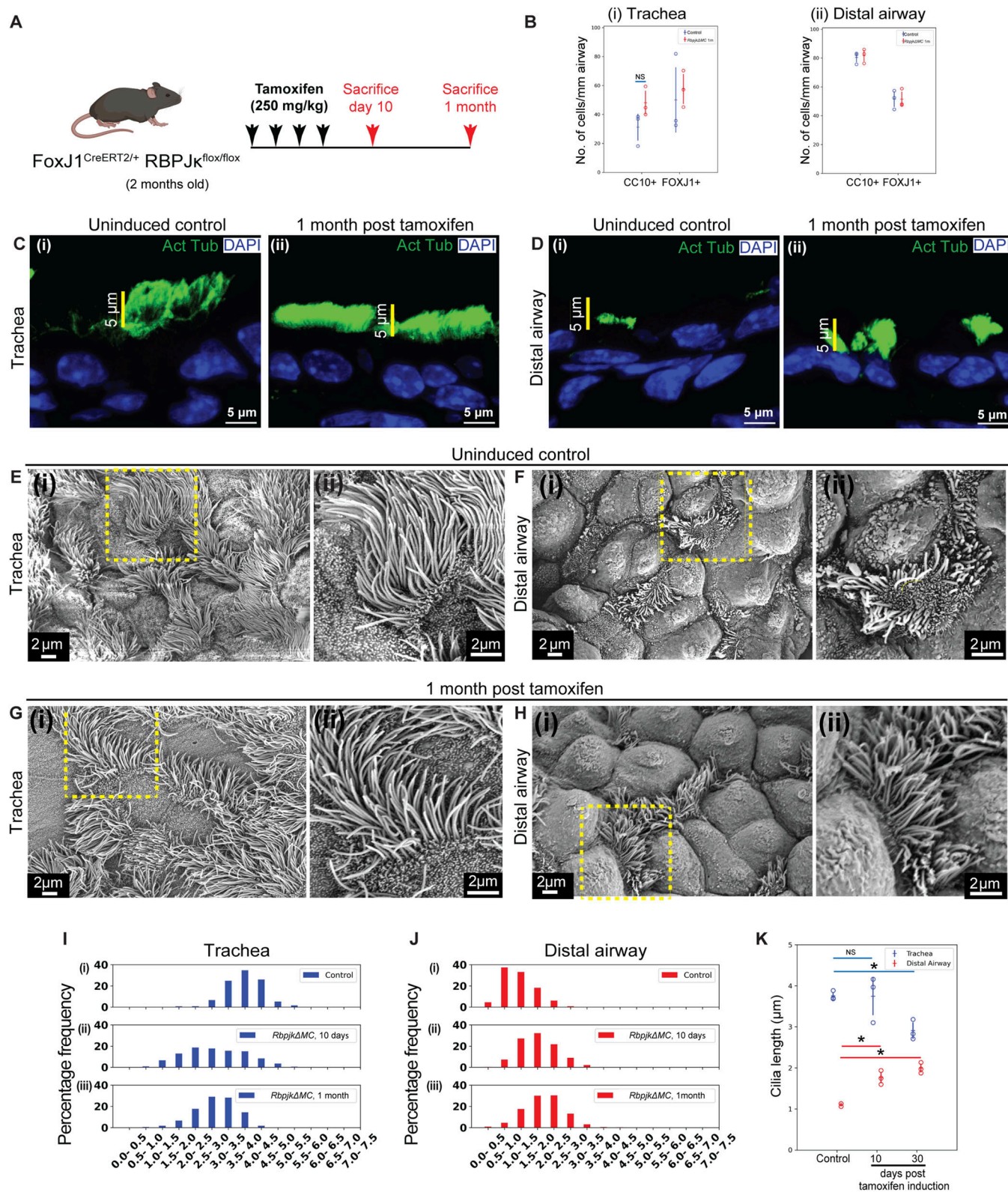

**Figure 2. Genetic ablation of *Rbpjk* in MCs alters ciliary lengths and the P-D gradient.**
**(A)** Outline of the genetic strategy used to conditionally delete *Rbpjk* in MCs. **(B)** Dot plots showing frequencies of club cells (CC10+) and MCs (FOXJ1+) in the trachea (B) (i) and distal airway (B) (ii) from control (uninduced, blue) and mutant (induced, "*RbpjkΔMC*," red) lungs. Hollow circles indicate mean cell frequency/mm airway in immunostained sections from an individual animal (see text, Materials and Methods section). Data in dot plot represent the mean ± SD (n = 3 animals). **(C, D)** Ciliary staining in the trachea (C) and distal airway (D) of control (age-matched uninduced littermate, (i)) and *RbpjkΔMC* (ii) lungs. **(C, D)** (i) Distribution of anti-acetylated tubulin (green) in airways from control lungs. **(C, D)** (ii) Distribution of anti-acetylated tubulin (green) in airways from *RbpjkΔMC* lungs. Note the pattern of ciliary

expression of ciliogenesis-related genes. A curated list of 402 "ciliogenesis" genes (GeneOntology.org, *Mus musculus*; Table S3) was analyzed for differential expression in *RbpjkΔMC* samples. Forty-four of these genes were altered in the trachea and five in the distal airway ($|\log_2FC| > 0.5$, $P < 0.05$; Fig S3F and G). No singular pathway that could directly explain the observed changes in ciliary length was identified. Thus, although the transcriptomics data confirm Notch-dependent regulation of spatial gene expression heterogeneity, the effect of these changes on ciliary length is likely to be multifactorial and complex.

Given the absence of a single ciliogenesis pathway explaining these changes, we next sought candidate regulators of ciliary length. Prominin-1 (*PROM1/CD133*), a transmembrane glycoprotein associated with microvilli and ciliary membranes, has been implicated in length regulation (Jászai et al, 2020; Serra et al, 2022). During development, Notch establishes a proximal–distal gradient of PROM1 expression in airway MCs, with highest levels distally; *Prom1* knockdown in air–liquid interface cultures increases ciliary length in a dose-dependent manner (Serra et al, 2022). Whether this relationship persists in adult MCs is unknown. Spatial transcriptomics analysis of adult lungs showed that *Prom1* mRNA levels form a distal-high gradient in controls (distal > trachea, $\log_2FC = 1.48$, $P = 0.01$). In *RbpjkΔMC* trachea, *Prom1* expression increased relative to controls ($\log_2FC = 0.5$, $P = 0.01$), consistent with the possibility that elevated PROM1 contributes to ciliary shortening. In contrast, *Prom1* expression in distal airways was unchanged ($\log_2FC = -0.18$, $P = 0.5$). These data are consistent with Notch-dependent regulation of PROM1 as a modulator of ciliary length.

To investigate this possibility, we examined PROM1 protein distribution in adult lungs by immunostaining coupled with signal amplification (see the Materials and Methods section). PROM1 colocalized with acetylated tubulin, confirming localization to MCs. PROM1 staining in adults was uniform across tracheal and distal airways (Fig 3A and B). In *RbpjkΔMC* lungs (1 mo post-tamoxifen), PROM1 levels were comparable to controls in tracheal segments but reduced in the distal airways (Fig 3C and D). Similarly, anti-Jagged1/2 antibody treatment caused progressive PROM1 loss in distal airways—detectable at 7 d and more pronounced by 1 mo (Fig 3E–H). The consistent reduction of PROM1 after Notch inhibition suggests that the Notch–PROM1 axis regulates ciliary length. These findings are consistent with a model in which Notch signaling maintains ciliary architecture, at least in part, by sustaining PROM1 expression in multiciliated cells.

## *Mycobacterium tuberculosis* infection induces multiciliated cell remodeling in distal airways similar to Notch inhibition

Having established that Notch signaling in multiciliated cells (MCs) regulates ciliary structure and airway homeostasis, we next examined how this regulatory axis responds to environmental challenges. Because the mucociliary escalator clears both commensal and pathogenic material, we reasoned that microbial status might influence Notch-dependent control of MC homeostasis. Two models were analyzed: germ-free mice lacking an intrinsic microbiome, and mice infected with *M. tuberculosis* (*M. tb*, H37Rv strain).

Ciliary organization in germ-free lungs was indistinguishable from that in specific pathogen–free controls (Fig S7), indicating that the native airway microbiota does not affect MC differentiation or maintenance.

In contrast, *M. tb* infection, which triggers chronic pulmonary inflammation and granuloma formation (Guirado & Schlesinger, 2013; Nemeth et al, 2020), caused marked structural remodeling of MCs in distal airways. After aerosolized infection, bacterial colony-forming unit (CFU) counts in lungs peaked by 1 mo and remained stable thereafter (Fig 4B). Lungs were analyzed 1 d, 1, 3, and 6 mo postinfection for bacterial load, ciliary morphology, and gene expression (Fig 4A; see the Materials and Methods section).

Immunostaining for acetylated tubulin and SEM revealed no change in tracheal cilia at any time point (Figs 4C (i–iii) and S8A). However, by 3 mo postinfection, distal airway MCs displayed a striking increase in ciliary length (Figs 4D (i–iii) and S8B), resembling the phenotype observed in Notch-deficient mice. Quantitative SEM analysis confirmed that tracheal ciliary length remained constant across time points (Fig 4E (i), G (i), and I (i); quantified in Fig 4K), whereas distal airway cilia were significantly longer by 3 mo postinfection (Fig 4F (i), H (i), and J (i); quantified in Fig 4K). Notably, this elongation occurred uniformly across distal airways and was not spatially associated with nearby granulomas.

These observations indicate that chronic *M. tb* infection elicits distal airway remodeling characterized by ciliary elongation, a phenotype analogous to that produced by Notch signaling inhibition. This suggests that sustained inflammatory or infectious stress can perturb Notch-mediated regulation of multiciliated cell homeostasis.

staining in the trachea and distal airway in control and the absence of this pattern in *RbpjkΔMC*, 1 mo post-induction. **(E, F, G, H)** Scanning electron microscopy of sections from control (age-matched uninduced littermate) and *RbpjkΔMC* lungs. **(E, F)** Micrographs of trachea (E) (i, ii) and distal airway (F) (i, ii) from control lungs. Magnified images of boxed regions in (i) are shown in (ii). **(G, H)** Micrographs of trachea (G) (i, ii) and distal airway (H) (i, ii) from *RbpjkΔMC* lungs. Magnified images of boxed regions in (i) are shown in (ii). **(I, J, K)** Quantification of ciliary length in micrographs ($\mu$m; see the Materials and Methods section). **(I, J)** Frequency distribution of ciliary lengths in the trachea in control and *RbpjkΔMC* lungs. **(I)** (i–iii) Frequency distribution of ciliary lengths ($\mu$m) in the trachea (blue) in control (age-matched uninduced littermate, (i), n = 667 cilia), 10 d post-induction ((ii), n = 535 cilia) and 1 mo post-induction ((iii), n = 685 cilia). **(J)** (i–iii) Frequency distribution of ciliary lengths ($\mu$m) in the distal airway (red) in control (age-matched uninduced littermate, (i), n = 1,795 cilia), 10 d post-induction ((ii), n = 733 cilia) and 1 mo post-induction ((iii), n = 1,173 cilia). Note the alteration of ciliary lengths and reduction of the P-D gradient in *RbpjkΔMC* lungs. **(K)** Dot plots of ciliary lengths in the trachea (blue) and distal airway (red) in control (age-matched uninduced littermate), and *RbpjkΔMC* lungs 10 d and 1 mo post-tamoxifen induction. Hollow circles indicate mean ciliary length in individual animals. Data in dot plot represent the mean ± SD (n = 3 animals) (NS, nonsignificant) (* denotes $P < 0.05$, *t* test). See also Figs S4 and S5.

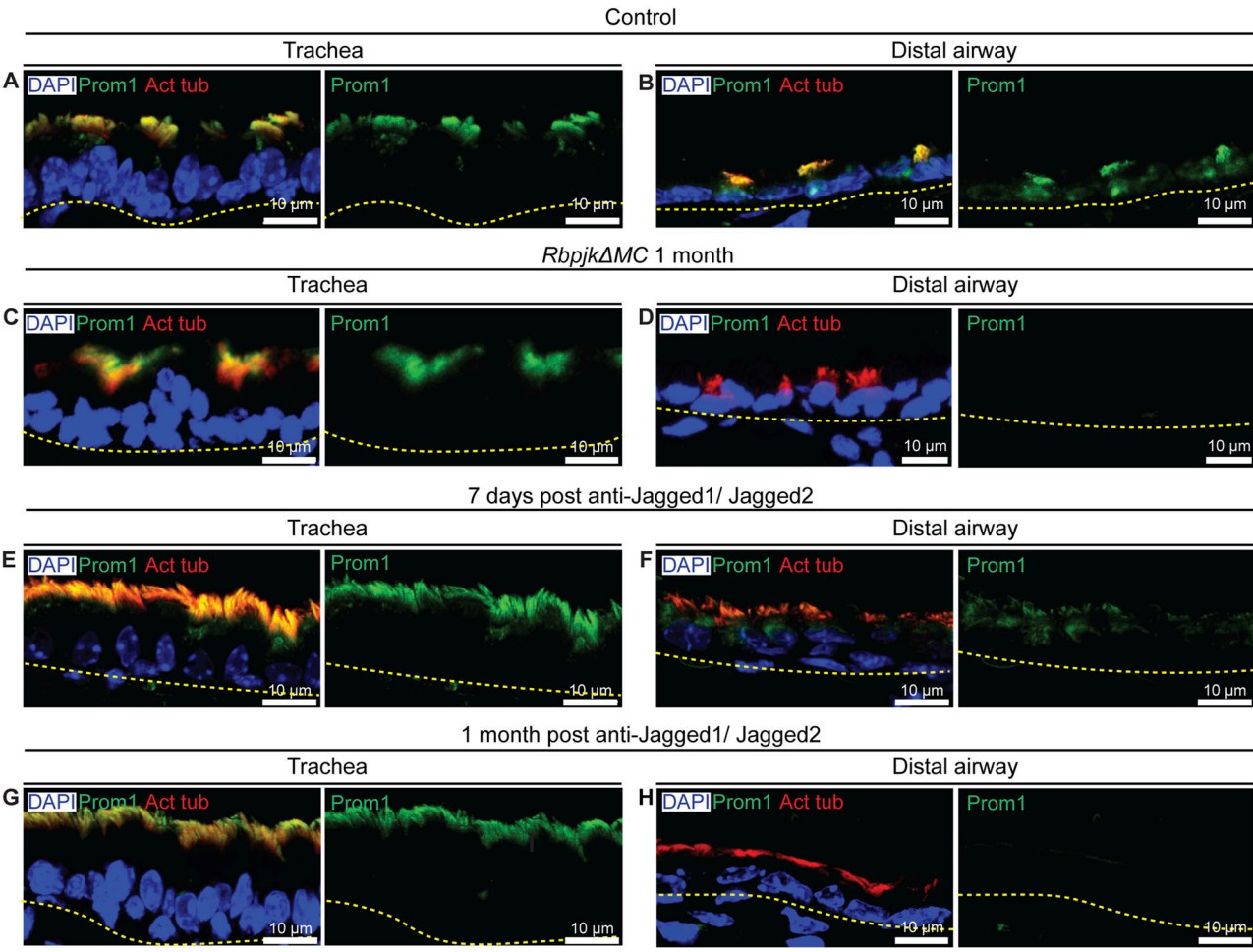

**Figure 3. Notch signaling controls the levels of prominin-1 expression in adult airway multiciliated cells.**
**(A, B)** Pattern of prominin-1 (PROM1) expression in multiciliated cells of trachea (A) and distal airways (B) in control adult lung tissues. Note that there is no obvious difference in the levels of PROM1 in the trachea and distal airway MCs. **(C, D)** Immunostaining for PROM1 in the trachea (C) and distal airways (D) of *RbpjkΔMC* animals 1 mo post-tamoxifen induction. Note the down-regulation of PROM1 in distal airway MCs compared with tracheal in 1-mo post–tamoxifen-treated samples. **(E, F)** Immunostaining for PROM1 in the trachea (E) and distal airways (F) of 7-d post–anti-Jagged1/Jagged2-treated animals. **(G, H)** PROM1 expression in the trachea (G) and distal airways (H) of 1-mo post–anti-Jagged1/Jagged2-treated animals. Note the down-regulation of PROM1 in distal airway MCs compared with tracheal MCs in anti-Jagged1/Jagged2-treated animals. **(A, B, C, D, E, F, G, H)** showing MCs stained with anti-acetylated tubulin (red) and PROM1 (green). Nuclei are stained with DAPI (blue). Yellow dotted lines indicate the basal margins of the airway epithelium.

## Alterations in multiciliated cell homeostasis during *M. tuberculosis* infection correlate with down-regulation of Notch signaling

Next, we investigated whether remodeling of distal airway cilia observed after *M. tb* infection correlates with perturbations in Notch signaling in MCs. To test this possibility, we assessed Notch pathway activity in infected lungs by immunostaining for the Notch2 intracellular domain (N2ICD), a marker of canonical Notch activation.

In uninfected controls, MCs in both tracheal and distal airways consistently exhibited nuclear N2ICD at all time points examined. In contrast, in *M. tb*-infected lungs, N2ICD localization was maintained in tracheal MCs but progressively lost in distal airway MCs. By 3 mo postinfection, nuclear N2ICD staining was markedly reduced or absent in the distal airways (Fig 5A–F), indicating a regional down-regulation of Notch signaling.

To complement these observations, we performed spatial transcriptomics profiling of tracheal and distal airway sections from mice 3 mo postinfection using the GeoMx Digital Spatial Profiler (10 tracheal and 12 distal airway ROIs; Fig 5G). Differential expression analysis ($|\log_2 FC| > 0.5$, $P < 0.05$) identified numerous region-enriched genes (372 tracheal, 443 distal airways; Table S1). We next examined the expression of curated Notch pathway target genes (from the MSigDB, REACTOME, and KEGG databases) relative to age-matched $FoxJ1^{CreERT2/+}$; $Rosa^{Tdtomato/+}$ controls. Expression of many Notch target genes—including *Notch2* itself—was significantly reduced in distal airway ROIs of infected lungs (Fig 5H). Together, these findings suggest that chronic *M. tb* infection leads to down-regulation of canonical Notch signaling specifically in distal airway MCs, coinciding with ciliary elongation.

Because PROM1 (CD133) is a Notch-regulated determinant of ciliary length, we next examined its expression in infected lungs.

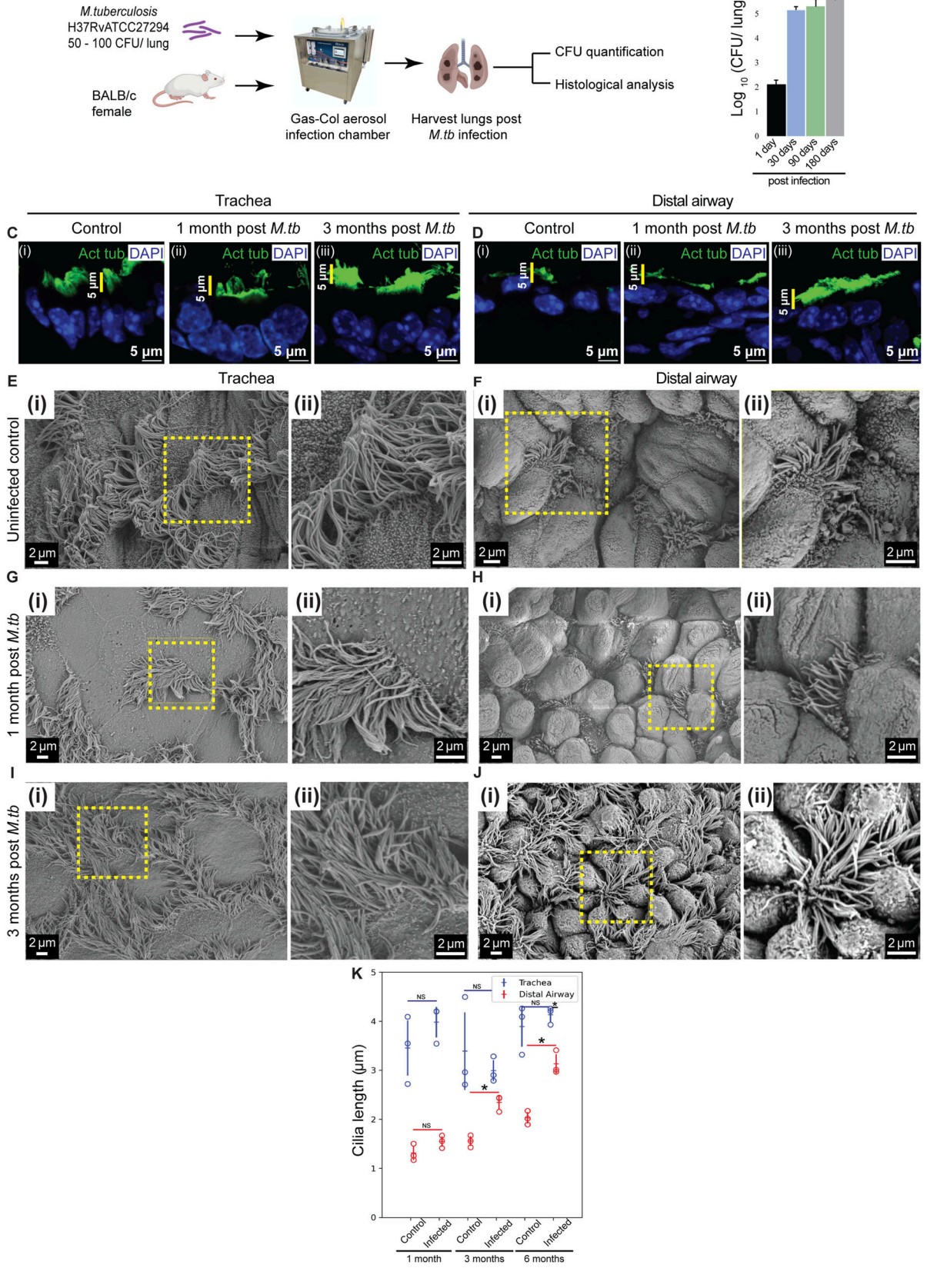

Immunostaining for PROM1 and acetylated tubulin in uninfected controls revealed colocalization in MCs and uniform staining intensity along the airway (Fig 5I and J). In contrast, PROM1 levels were visibly reduced in distal airways of lungs harvested 3 mo postinfection, whereas tracheal expression remained relatively unchanged (Fig 5K and L). This reduction paralleled the loss of N2ICD staining in the same regions.

These results indicate that chronic *M. tuberculosis* infection suppresses Notch signaling and down-regulates PROM1 expression in distal airway MCs. The concomitant elongation of distal cilia suggests that disruption of the Notch–PROM1 regulatory axis contributes to infection-induced remodeling of the multiciliated epithelium.

# Discussion

This study reveals that Notch signaling integrates with ciliary architecture and environmental cues to regulate the P-D ciliary gradient. We show that loss of Notch signaling—either through targeted pathway inhibition or after *M. tuberculosis* infection—alters ciliary length and disrupts the proximal–distal (P-D) gradient across the airways. These findings demonstrate that ciliary length and the P-D gradient are not passively maintained but are under active control, with Notch signaling serving as a key stabilizing mechanism.

The Notch pathway has established stage-specific roles in MC development and renewal (Tsao et al, 2009; Morimoto et al, 2010). Down-regulation of signaling in uncommitted progenitors by miR-34/449 initiates multiciliogenesis (Marcet et al, 2011), whereas later activation of the pathway refines MC differentiation. Recent work indicates that Notch contributes to establishing a P-D gradient of prominin-1 (PROM1/CD133) expression in committed MCs (Serra et al, 2022). PROM1 restricts ciliary growth in a dose-dependent manner, thereby defining regional differences in ciliary length. Our findings extend these observations to the adult lung: our findings indicate that PROM1 may act downstream of Notch signaling to regulate ciliary architecture. Inhibition of Notch signaling, either genetically or pharmacologically, resulted in down-regulation of PROM1 protein in distal airways, correlating with ciliary elongation. Similarly, *M. tuberculosis* infection led to a reduction in Notch activity and PROM1 expression in distal MCs, accompanied by ciliary remodeling. Together, these results suggest that the Notch–PROM1 axis remains active in adult airways to maintain

ciliary homeostasis. Although the parallel reduction of PROM1 and ciliary remodeling suggests a mechanistic connection, direct evidence for causality will require targeted manipulation of PROM1 levels in adult multiciliated cells.

The precise mechanism by which Notch–PROM1 signaling can modulate ciliary length is unclear. Studies in Madin–Darby canine kidney (MDCK) cells indicate that *PROM1* interacts with the ciliary depolymerization factor ADP-ribosylation factor–like 13B (Arl13b) to regulate microtubule turnover (Jászai et al, 2020), suggesting a potential molecular link. Future work should clarify whether similar interactions occur in airway MCs and whether Notch signaling directly regulates PROM1 turnover or localization. Moreover, the basis for the differential effects of Notch perturbation on PROM1 expression in proximal versus distal airways is currently unclear, raising the possibility that additional signaling pathways modulate PROM1 expression in a region-specific manner.

Our results also reveal that chronic infection and inflammation can alter Notch-dependent homeostasis. *M. tuberculosis* infection produced a distal airway phenotype resembling Notch inhibition, consistent with literature showing that alveolar injury and macrophage infiltration down-regulate Notch signaling (Choi et al, 2020, 2021). Bleomycin-induced lung injury, which elevates IL-1$\beta$ expression, reduces Jagged1/Jagged2 ligand availability through IL-1R1 signaling in MCs and consequently suppresses Notch activity (Choi et al, 2021). This macrophage–cytokine axis may similarly contribute to the down-regulation of Notch signaling observed in *M. tuberculosis* infection. Thus, inflammatory cues may modulate ciliary structure indirectly by repressing Notch–PROM1 signaling in distal airways.

Evidence from human pathology supports a broader relevance of this regulatory axis. Analyses of airway epithelia from smokers, asthmatic patients, and individuals with chronic obstructive pulmonary disease (COPD) show shortened cilia (Leopold et al, 2009; Thomas et al, 2010; Hessel et al, 2014; Robinot et al, 2021), whereas nasal multiciliated cells in patients with nasal polyps exhibit elongated cilia (Li et al, 2014). We propose that alterations in Notch signaling could underlie this spectrum of phenotypes in human airways.

Ciliary beating is known to be dynamically regulated by chemical and mechanical cues in the trachea and upper airways (Hayashi et al, 2005; Delmotte & Sanderson, 2006). Our findings identify an additional level of structural control, in which Notch signaling maintains the architecture of the mucociliary escalator itself. We suggest that modulation of ciliary length and the P-D gradient represents an adaptive

**Figure 4. *M. tuberculosis* infection induces multiciliated cell remodeling akin to Notch inhibition in the distal airway.**
**(A)** Experimental design for *M. tuberculosis* (*M. tb*) infection in BALB/c mice. **(B)** Frequencies of colony-forming units (CFU) of *M. tb* in infected lungs at 1 d, and 1, 3, , and 6 mo postinfection (n = 3 animals for each time point) **(C, D)** Ciliary staining in the trachea (C) and distal airway (D) of control (BALB/c uninfected age-matched cohort, (i)) and *M. tb*-infected lungs (ii, iii). **(C, D)** (i) Distribution of anti-acetylated tubulin (green) in airways from control lungs. **(C, D)** (ii) Distribution of anti-acetylated tubulin (green) in airways from *M. tb* 1 mo postinfection. **(C, D)** (iii) Distribution of anti-acetylated tubulin (green) in airways from *M. tb* 3 mo postinfection. Note the pattern of ciliary staining in the trachea and distal airway in controls and the absence of this pattern 3 mo post-*M. tb* infection. **(E, F, G, H, I, J)** Scanning electron microscopy of sections from control and *M. tb*-infected lungs. **(E, F)** Micrographs of trachea (E) (i, ii) and distal airway (F) (i, ii) from control lungs. Magnified images of boxed regions in (i) are shown in (ii). **(G, H)** Micrographs of trachea (G) (i, ii) and distal airway (H) (i, ii) from lungs 1 mo post-*M. tb* infection. Magnified images of boxed regions in (i) are shown in (ii). **(I, J)** Micrographs of trachea (I) (i, ii) and distal airway (J) (i, ii) from lungs 3 mo post-*M. tb* infection. Magnified images of boxed regions in (i) are shown in (ii). **(K)** Dot plot of ciliary lengths in the trachea (blue) and distal airway (red) from control lungs (uninfected, age-matched, n = 314 in trachea, n = 397 in distal airway) and from lungs 1 mo (n = 353 in trachea, n = 645 in distal airway), 3 mo (n = 356 in trachea, n = 1,082 in distal airway), and 6 mo (n = 230 in trachea, n = 1,079 in distal airway; see Fig S8) post-*M. tb* infection. Hollow circles indicate mean ciliary length in individual animals. Data in dot plot represent mean ± SD (n = 3 animals) (NS, nonsignificant) (* denotes $P < 0.05$, *t* test). See also Figs S7 and S8.

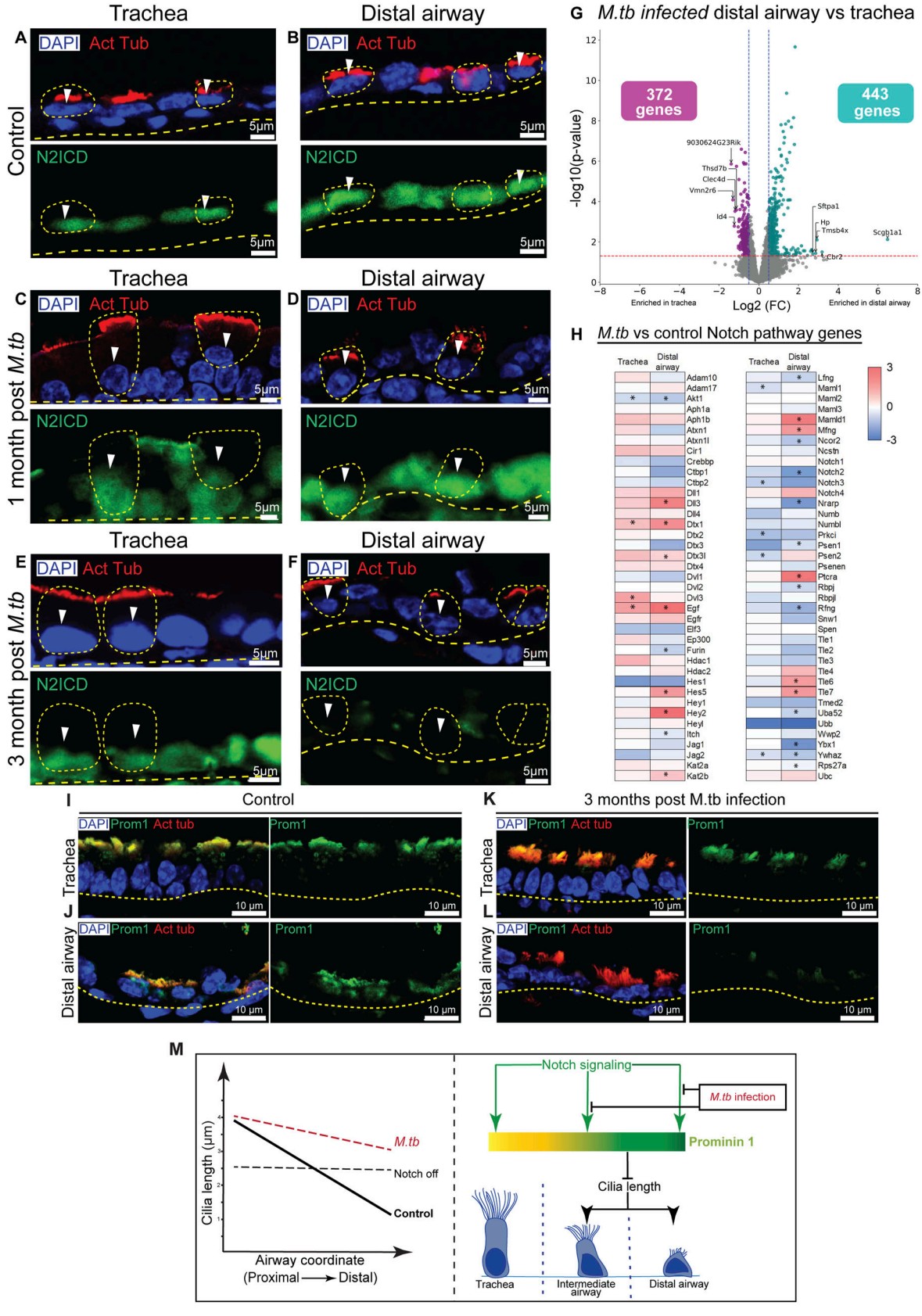

mechanism that tunes mucociliary transport and airway clearance efficiency. Active regulation of ciliary structure may therefore be a general feature of ciliated epithelia, ensuring unobstructed airflow across developmental, physiological, and environmental contexts.

# Materials and Methods

### Study design

The main aim of the study is to elucidate the role of Notch signaling in MCs during homeostasis and upon pathogen challenge. We inhibited Notch signaling using two approaches: first, a systemic inhibition using inhibitory antibodies against ligands for Notch receptors (Jagged1 and Jagged2); and second, using a genetic approach by conditionally deleting *Rbpjk*, in *Foxj1*-expressing MCs and investigating the transcriptional changes using a spatial transcriptomics approach. We carefully looked at the ciliary architecture and P-D ciliary gradient in different regions of the airways. A murine model of *M. tb* infection was used to probe the impact of pathogen challenge in ciliary remodeling.

### Animal ethics and animal handling

All the experiments involving mice were conducted at either inStem, Bangalore, or FNDR (Foundation for Neglected Disease Research), Bangalore, or at the National Gnotobiotic Rodent Resource Center (NGRRC) at the University of North Carolina (UNC), Chapel Hill. For experiments conducted at inStem and FNDR, the procedures were reviewed and approved in advance by the Institutional Animal Ethical Committee in accordance with guidelines established by the Committee for Control and Supervision of Experiments on Animals (CCSEA) (FNDR registration number—2082/PO/Rc/S/2019/CCSEA). For experiments conducted at UNC, the studies and protocols were approved by the Institutional Animal Care and Use Committee for the University of North Carolina in accordance with the guidelines outlined by the Animal Welfare and the National Institutes of Health.

Groups of 3–5 age-matched adult animals that were 6–8 wk of age were used for all the experiments. Female BALB/c animals were used for all *M. tb* infection experiments. For all other experiments, animals of either gender, males or females, were used depending on the available genotype. Any procedure that could conceivably cause distress to the animals employed periprocedural anesthesia with isoflurane gas (Baxter Healthcare Corp.) delivered by an esthetic vaporizing machine. In addition, all animals were monitored for signs of distress and euthanized if in distress. Euthanasia was performed by anesthetizing the animals followed by cervical dislocation.

### Mouse strains

All animal strains were housed under specific pathogen–free (SPF) environment conditions in the respective animal facilities at the Bangalore Life Science Cluster, FNDR, and NGRRC. C57BL/6J (JAX# 000664), *FoxJ1*<sup>CreERT2</sup> (JAX# 27012), BALB/c, and Rosa26 Ai14 Tdtomato (JAX# 007914) were obtained commercially from the Jackson Laboratory. Floxed *RBP-J* strain (RIKEN# RBRC01071) was a kind gift from Dr. Mitsuru Morimoto, RIKEN. Frozen embryos of the Floxed *RBP-J* strain were cryo-rederived at the Mouse Genome and Engineering Facility at the National Center for Biological Sciences. The *FoxJ1*<sup>CreERT2/+</sup> strain was crossed to Floxed *RBP-J* and Rosa26 Ai14 Tdtomato strains to generate *FoxJ1*<sup>CreERT2/+</sup>; *Rbpjk*<sup>flox/flox</sup> and *FoxJ1*<sup>CreERT2/+</sup>; Rosa<sup>Tdtomato/+</sup> strains, respectively.

Genotyping of the mouse strains was performed using the following primers:

#### *FoxJ1*<sup>CreERT2</sup>

5′-GCAGATGGAGAGAGGTGGAG-3′
5′-CTTGGCGTTGAGAATGGAGA-3′
5′-ATTGCATCGCATTGTCTGAG-3′

#### *Floxed RBP-J*

5′-CTGAGTAAGATGAGATGCTGACATCTGA-3′
5′-ATGTACATTTTGTACTCACAGAGATGGATG-3′
5′-GCTTGAGGCTTGATGTTCTGTATTGC-3′

---

**Figure 5. *M. tuberculosis* infection perturbs Notch signaling in the distal airway.**
**(A, B, C, D, E, F)** Status of Notch signaling in MCs in uninfected and *M. tb*-infected lungs. **(A, C, E)** Tracheal sections from control (BALB/c age-matched uninfected cohort, (A)), 1 mo post-*M. tb* infection (C) and 3 mo post-*M. tb* infection (E) showing MCs (white arrowheads, demarcated by yellow dotted circles) stained with anti-acetylated tubulin (red, upper panels) and N2ICD (green, lower panels). Nuclei are stained with DAPI (blue). Yellow dotted lines indicate the basal margin of the airway epithelium. **(B, D, F)** Distal airway sections from control (BALB/c age-matched uninfected cohort, (B)), 1 mo post-*M. tb* infection (D) and 3 mo post-*M. tb* infection (F) showing MCs (white arrowheads) stained with anti-acetylated tubulin (red, upper panels) and N2ICD (green, lower panels). Nuclei are stained with DAPI (blue). Yellow dotted lines indicate the basal margin of the airway epithelium. Note that nuclear N2ICD staining is absent in MCs of distal airway in 3 mo post–*M. tb*-infected lungs. **(G)** Volcano plot comparing gene expression in tracheal and distal airway ROIs in lungs 3 mo post-*M. tb* infection. Individual genes (dots) showing log2 fold change > 0.5, *P* < 0.05 (enriched in distal airway, cyan), and log2 fold change < −0.5, *P* < 0.05 (enriched in trachea, purple) are highlighted. The number of highlighted genes in the trachea and distal airway is shown in boxes in corresponding regions of the volcano plot. Top five enriched genes in the trachea and distal airway are labeled. **(H)** Heat map showing the differential expression of genes involved in Notch signaling (MSigDB REACTOME database and KEGG database combined) in the trachea and distal airway of *M. tb*-infected animals compared with the control. Note the decrease in the expression of the Notch pathway genes and Notch2 in particular in the distal airway postinfection. Genes showing a statistically significant change are marked with asterisk (*). **(I, J)** Pattern of PROM1 expression in multiciliated cells of trachea (I) and distal airways (J) in control adult lung tissues. Note that there is no obvious difference in the levels of PROM1 in the trachea and distal airway MCs. **(K, L)** Immunostaining for PROM1 in the trachea (K) and distal airways (L) of 3-mo post–*M. tb*-infected animals. Note the down-regulation of PROM1 in distal airway MCs compared with tracheal in *M. tb*-infected lung. **(I, J, K, L)** showing MCs stained with anti-acetylated tubulin (red) and PROM1 (green). Nuclei are stained with DAPI (blue). Yellow dotted lines indicate the basal margin of the airway epithelium. **(M)** Graphical summary of the perturbations in ciliary lengths and the P-D gradient reported here and a model for the role of Notch signaling in MCs in stabilizing ciliary length.

---

***Rosa***<sup>*Tdtomato*</sup>

 5'-AAGGGAGCTGCAGTGGAGTA-3'
 5'-CCGAAAATCTGTGGGAAGTC-3'
 5'-CTGTTCCTGTACGGCATGG-3'
 5'-GGCATTAAAGCAGCGTATCC-3'

## Antibody-mediated Notch inhibition

C57BL/6 mice and *FoxJ1*<sup>CreERT2/+</sup>; *Rosa*<sup>Tdtomato/+</sup> mice (≥6 wk of age) were injected intraperitoneally with nontargeting IgG (anti-gp120) or anti-Notch2/Notch1 or anti-Jagged1/Jagged2 antibodies diluted in PBS and euthanized at the time points indicated in the experimental schematics. Antibody concentrations used were as follows: nontargeting IgG (anti-gp120), 40 mg/kg; anti-Notch1, 10 mg/kg; anti-Notch2, 30 mg/kg; anti-Jagged1, 20 mg/kg; and anti-Jagged2, 20 mg/kg. Animals were euthanized at respective time points as indicated in figures.

## Genetic ablation of *Rbpjk* in multiciliated cells

*FoxJ1*<sup>CreERT2/+</sup>; *Rbpjk*<sup>flox/flox</sup> mice (6–8 wk of age) were injected with tamoxifen (250 mg/kg body weight; Sigma-Aldrich) four times on alternate days before euthanizing at the time points indicated in the figures.

## Infection with *M. tuberculosis* (*M. tb*)

6- to 8-wk-old female BALB/c animals were infected with H37Rv strain of *M. tuberculosis* (ATCC27294) in a Glas-Col Aerosol Infection Chamber (Glas-Col), calibrated to deliver ~100 to 200 colony-forming units of *M. tb*, in a BSL-3 facility. Infected mice were housed in IVC isolators (Citizen, India) during the entire period of experimentation. Lungs of infected animals (n = 3 for CFU and n = 5 for histology) were collected along with the healthy/naive mouse controls (n = 3) on 1, 30, 90, and 180 d postinfection.

## Germ-free (GF) mice

GF mice were generated in the C57BL/6J background at the CGBID Gnotobiotic Core, UNC. These mice had no detectable levels of yeast, bacteria, parasites, or molds. The sterility of the isolators housing the GF mice was regularly monitored. Samples from feces, buccal–paws–cage swabs, food, and drinking water were regularly examined for bacterial contamination through Gram staining and 16S PCR on sampled feces.

## Histology, immunofluorescence, and imaging

Lungs were inflated with 4% (weight/volume) PFA + 2% low-melting agarose in PBS and immersed in the same fixative for 12 h at 4°C. The lobes were separated, and the left lobe was used for making thick sections (200 μm). The remaining lobes were incubated in PBS at 60°C overnight for melting the agarose and subsequently processed for embedding into paraffin for histological analysis. Heat-mediated antigen retrieval was performed using antigen unmasking solution from Vector Laboratories (pH 9) before immunostaining. pH 6 antigen unmasking solution was used when

staining for Notch2ICD. Immunofluorescence analysis used the following antisera: rabbit anti-Notch2 (1:100; Abcam), rat anti-prominin-1 (1:100; Sigma-Aldrich), goat anti-uteroglobin (1:1,000; Merck), mouse anti-acetylated tubulin (1:2,000; Sigma-Aldrich), mouse anti-Foxj1 (1:250; eBioscience), rabbit anti-RBPUSH (1:200; Cell Signaling Technologies), mouse anti-RFP (1:300; Abcam), rabbit anti-RFP (1:300; Rockland), rabbit anti-acetylated tubulin (1:1,000; Abcam). For N2ICD and prominin-1 after primary antibody incubation, sections were washed and incubated with respective biotinylated secondary antibodies (1:200; Jackson Laboratories) followed by tyramide signal amplification: Alexa 405/488/568/647–conjugated donkey anti-mouse/rabbit/goat (1:300; Invitrogen). Sections were imaged using 20X (NA-0.8) or 63X (NA-1.4) objectives either on Zeiss LSM 780 or on LSM 980 laser-scanning confocal microscopes.

## Quantification of cell frequencies

Cell frequencies were calculated from immunostained paraffin sections (4–5 μm). Multichannel images were acquired using confocal microscopes, and cell frequencies were obtained by manual counting in images using Fiji (ImageJ). Nuclei were identified by the presence of DAPI, and cells that were positive for the respective markers were counted. For CCs and MC frequencies, CC10 and FOXJ1 immunostained cells were quantified and normalized to the length of basal lamina both in the trachea and in the distal airway. Closed airways with a diameter of ≤200 or 200 μm from the bronchoalveolar duct junctions (BADJ) of open-ended airways that transition into the alveoli were considered as distal airways. All frequencies are plotted as the mean ± SD from n = 3 animals per condition. A *t* test was used for calculating the significance in the statistical analysis.

## SEM

The left lobe from each of the animals was sectioned (200 μm) using a Compresstome. Sections were washed repeatedly in phosphate buffer at 60°C for 10 min, washed to melt the agarose, and fixed with 2.5% glutaraldehyde overnight at 4°C. Fixed lung sections were dehydrated with increasing concentrations of acetone and dried by critical point drying using Leica EM CPD 300. Sections were then sputter-coated with gold particles using PELCO SC-7. Electron micrographs of the lung sections were taken using the ZEISS MERLIN Compact at a magnification of 3–10 kX and an accelerator potential of 2–4 kV.

## Measurement of ciliary length

Linear measurements of individual cilia of MCs were measured manually from SEM images (Polino et al, 2023). Cilia whose entire length from base to the tip was clearly discernible were traced using the freehand option in Fiji (ImageJ), and their lengths were measured and tabulated. 200–1,000 cilia per animal per condition were measured across multiple SEM images (n ≥ 3 animals per condition). Graphs for ciliary lengths were generated in Microsoft Excel by compiling the results to calculate the mean and mean ± SD. A *t* test was used for calculating the *P*-value and significance. Closed airways with a diameter of ≤100 or 100 μm from the BADJ of

open-ended airways that transition into alveoli were considered for imaging cilia.

## Spatial transcriptomics

Formaldehyde-fixed, paraffin-embedded (FFPE) sections of trachea and lungs (5 μm) from control, *FoxJ1*$^{CreERT2/+}$; *Rbpjk*$^{flox/flox}$, and *M. tb*-infected mice were used for spatial transcriptomics profiling. Mouse anti-acetylated tubulin (Sigma-Aldrich) antibody was conjugated to CF568 using Mix-n-Stain CF 568 Antibody Labeling Kit (50–100 μg; Sigma-Aldrich). Double positivity to acetylated tubulin and SYTO 13 (nuclear marker) was used to identify the MC-containing airway epithelium and define ROIs. GeoMx Digital Spatial Profiler (DSP) slide preparation was performed as per the manufacturer's instructions. Slides were then incubated overnight at 37°C with Mouse Whole Transcriptome Atlas (WTA) probes (NanoString) as per the NanoString GeoMx RNA-NGS manual instructions. Slides were then washed, stained with acetylated tubulin–CF568 and SYTO 13 for 1 h at RT, and imaged on the GeoMx DSP at 20X magnification, and ROIs were selected in the trachea and the distal airway. Indexed oligonucleotides from each ROI were released and deposited into a 96-well plate on exposure to 385-nm light (UV), with a microcapillary tube. The collected DNA oligos from each ROI were subjected to Illumina library preparation for sequencing. A total of 74 ROIs and 2 NTC were profiled. Sequencing libraries were prepared according to the NanoString GeoMx-NGS Readout Library Prep manual. The PCR was performed using the NanoString SeqCode primer for 21 cycles. Libraries were sequenced on an Illumina NovaSeq 6000, with 2 × 151 bp paired-end reads.

## NanoString GeoMx WTA data analysis

The sequenced FASTQ files were processed further to convert into a .dcc (digital count conversion) format using GeoMx-NGS Pipeline v.2.3.3.10 from NanoString. The trimming of adapters, removal of duplicates and mapping to the reference probe barcodes present in the WTA panel, and quantification would be performed during the conversion process of FASTQ to dcc files. The individual ROIs would hence result in a single dcc file. The resulting dcc files from each AOI were further imported and analyzed using the GeoMxTools (1) R package. R version 4.2 (https://www.r-project.org/about.html) was used throughout the analysis unless stated otherwise. The analysis within the GeoMxTools pipeline includes QC for both Probe and Segment, Normalization, Unsupervised clustering. The parameters used during the QC of segments include the following: minSegmentReads = 1,000; percentTrimmed = 80; percentStitched = 80; percentAligned = 80; percentSaturation = 50; minNegativeCount = 10; maxNTCCount = 1,000; minNuclei = 200; minArea = 5,000; and minLOQ = 2. The parameters used for the QC or filtering of probes (target genes) include the following: geometric mean of each probe's counts from all segments divided by the geometric mean of all probe counts representing the target from all segments <0.1 and percentage of segments within which the probe is identified as an outlier based on Grubb's test ≥ 20%. A probe is removed locally (from a given segment) if the probe is an outlier according to Grubb's test in that segment.

After appropriate QC on segments and target genes, the counts were normalized using the Q3 method of normalization. The differential gene expression testing was performed using the LMM within the GeoMxTools R package, which adjusts for the fact that the multiple ROIs placed per tissue section are not independent observations as understood by other statistical tests.

## Statistical analysis

A two-tailed unpaired *t* test was employed to determine the statistical significance between conditions in the measurements of ciliary length and frequencies. *P*-value ≤ 0.05 is considered statistically significant and is denoted by "*" in the corresponding plot. All experiments were conducted independently on at least three animals (n ≥ 3), and error bars indicate the mean ± SD. Matplotlib (python 3.0 package) was used to plot dot plots and frequency distribution.

## Online supplemental material

Fig S1 shows the ciliary length along the P-D axis of the airway epithelium. Fig S2 shows ciliary remodeling 7d post–anti-Jagged1/2 treatment. Fig S3 shows the fate of MCs and motile cilia 3 mo post–anti-Jagged1/Jagged2 treatment. Fig S4 shows efficiency of genetic ablation of RBPJκ in MCs. Fig S5 shows ciliary remodeling in 10-d post–tamoxifen-treated *RbpjkΔMC* animals. Fig S6 shows spatial transcriptomics results, suggesting that Notch signaling in MCs regulates airway gene expression and homeostasis. Fig S7 shows ciliary length in germ-free mice. Fig S8 shows ciliary morphology 6 mo post-*M. tb* infection.

# Data Availability

All data required to evaluate the conclusions are available in the main text or the supplementary materials. Raw data underlying this work are available from the corresponding author upon reasonable request.

# Supplementary Information

# Acknowledgements

The team expresses gratitude to TheraCUES Innovations Private Limited, Bangalore, for spatial transcriptomics experiment and Manju Moorthy for post-sequencing analysis. Electron Microscopy Facility, Central Imaging and Flow Cytometry Facility (CIFF), and the Animal Care and Resource Centre (ACRC) at the Bangalore Life Science Cluster (BLiSC) have provided their support in conducting experiments and maintaining experimental animals. A Guha would like to thank Dr. Balasubramanian V for inspiration, Dr. Madan Rao for discussions, and Dr. Narmada Khare for critical reading of the article. Funding for this study was obtained from the following sources: Institutional (inStem) core funds from Department of Biotechnology (DBT, A Guha),

Department of Science and Technology—Science and Engineering Research Board, India (A Guha), Chan Zuckerberg Initiative (Mapping the Paediatric Inhalation Interface: Nose, Mouth and Airways) (JS Hagood, RC Boucher, KM Byrd, and A Guha), Cystic Fibrosis Foundation Research Development Project BOUCHE19R0 (RC Boucher), National Institute of Health grants NHLBI P01HL164320 and NIDDK P30 DK065988 (RC Boucher), National Institute of Health grants NHLBI R01 HL 150541-01 (A Livraghi-Butrico), Council of Scientific and Industrial Research (CSIR) 09/860(1273)/2019-EMR-I (N Joy), and Fellowship from DBT (SM Lingamallu and A Deshpande).

## Author Contributions

N Joy: conceptualization, data curation, formal analysis, validation, investigation, visualization, methodology, and writing—original draft, review, and editing.
A Deshpande: formal analysis and investigation.
SM Lingamallu: formal analysis and investigation.
VM Prabantu: data curation and formal analysis.
CN Naveenkumar: resources, formal analysis, and investigation.
K Bharathkumar: resources and investigation.
S Bhat: formal analysis and investigation.
Z Alvarado-Martinez: formal analysis.
A Livraghi-Butrico: resources and investigation.
JS Hagood: resources and investigation.
RC Boucher: resources and investigation.
D Lafkas: resources.
KM Byrd: conceptualization, resources, and investigation.
S Narayanan: resources and investigation.
RK Shandil: resources and investigation.
A Guha: conceptualization, resources, formal analysis, supervision, funding acquisition, visualization, project administration, and writing—original draft, review, and editing.

## Conflict of Interest Statement

The authors declare that they have no conflict of interest.

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
