## [Reviewer comments · Life Science Alliance]

Notch signaling stabilizes lengths of motile cilia in multiciliated cells in the lung

Neenu Joy, Aditya Deshpande, Sai Manoz Lingamallu, Vasam Prabantu, Naveenkumar CN, Bharathkumar K, Sukanya Bhat, Zabdriel Martinez, Alessandra Livraghi-Butrico, James Hagood, Richard Boucher, Daniel Lafkas, Kevin Byrd, Shridhar Narayanan, R Shandil, and Arjun Guha

DOI: <https://doi.org/10.26508/lsa.202503268>

Corresponding author(s): Arjun Guha, Institute for Stem Cell Biology and Regenerative Medicine

Review Timeline:

Submission Date:	2025-02-14
Editorial Decision:	2025-04-16
Revision Received:	2025-10-29
Editorial Decision:	2025-12-02
Revision Received:	2025-12-18
Accepted:	2025-12-23

Scientific Editor: Sarita Hebbar

Transaction Report:

April 16, 2025

Re: Life Science Alliance manuscript #LSA-2025-03268-T

Dr. Arjun Guha Guha
Institute for Stem Cell Biology and Regenerative Medicine
NCBS, GKVK Campus
Bellary Road
Bengaluru, Karnataka 560064
India

Dear Dr. Guha,

Thank you for submitting your manuscript entitled "Notch signaling stabilizes lengths of motile cilia in multiciliated cells in the lung" to Life Science Alliance. The manuscript was assessed by three expert reviewers, whose comments are appended to this letter.

Overall, the three reviewers found this work of potential value to the field. That said, we concur with the reviewers that specific aspects of the manuscript need to be revised for publication at LSA, namely:

1. Details on the characterisation of ciliated cells labelled by N2ICD staining in terms of their cellular origin (basal cells staining, Reviewer 1, comment 1), the baseline staining, spatial differences (Reviewer 3, comment 3) and staining data for adult stages (Reviewer 2, comment 1). Please also include appropriate controls of your choice for this staining, as suggested by Reviewer 1.
2. Information on the deletion of Rbpjk in multi-ciliated cells including ciliary length data at 10days (Reviewer 2, comment 3) and efficiency of the deletion (Reviewer 3, comment 3).
3. Expand on the description of the spatial transcriptomics data following the suggestions of Reviewer 2 and 3, comment 4.
4. Details on results and discussion on ciliary length phenotype (Reviewer 1, comment 4, and Reviewer 3, last comment)

Given the reviewers' interest and recommendations, we would like to invite you to submit a revised manuscript addressing the reviewers' comments. The typical timeframe for revisions is three months.

When submitting the revision, please include a letter addressing all the reviewers' comments point by point. While a rebuttal must respond to all points in some form, additional data to resolve these points is not required. Please be aware that all data referred to in the manuscript must be included in the manuscript.

Please note that papers are generally considered through only one revision cycle, so strong support from the referees on the revised version is needed for acceptance. While you are revising your manuscript, please also attend to the editorial points, appended below, to help expedite the publication of your manuscript.

Thank you for this interesting contribution to Life Science Alliance. We are looking forward to receiving your revised manuscript.

Sincerely,

Sarita Hebbar, PhD
Scientific Editor
Life Science Alliance
<http://www.lsjournal.org>

- A letter addressing the reviewers' comments point by point.
- An editable version of the final text (.DOC or .DOCX) is needed for copyediting (no PDFs).
- High-resolution figure, supplementary figure and video files uploaded as individual files: See our detailed guidelines for preparing your production-ready images, <https://www.life-science-alliance.org/authors>
- Summary blurb (enter in submission system): A short text summarizing in a single sentence the study (max. 200 characters including spaces). This text is used in conjunction with the titles of papers, hence should be informative and complementary to the title and running title. It should describe the context and significance of the findings for a general readership; it should be written in the present tense and refer to the work in the third person. Author names should not be mentioned.
- By submitting a revision, you attest that you are aware of our payment policies found here: <https://www.life-science-alliance.org/copyright-license-fee>

B. MANUSCRIPT ORGANIZATION AND FORMATTING:

Reviewer #1 (Comments to the Authors (Required)):

In this manuscript the authors demonstrate that Notch signaling plays a critical role in regulating the length of cilia in ciliated cells of the airway epithelium. They show that the proximal-distal gradient in ciliary length is impacted by environmental challenges such as M. tuberculosis infection as well as by Notch perturbations performed by using neutralizing antibodies or genetic mouse lines. Thus concluding that Notch signaling serves as a stabilizing mechanism.

The authors clearly show an association of Notch changes and alterations in ciliary length but the mechanism remains unknown. In fact, one misses data directly demonstrating this association instead of so much speculation.

Major concerns raised are as follow:

1. The authors show N2ICD staining in ciliated cells. It was previously shown that N2ICD is only expressed in secretory (Club) cells in the tracheal epithelium (PMID: 26147083) and in the distal airway epithelium (PMID: 26580007) at homeostasis. Since this is contrary with what was previously reported, and the images only show few cells (in fact it seems that all cells are positive. Are there basal cells in this tracheal image (Fig.1b)?), it is required to demonstrate that this is real staining with multiple controls (non-primary, IgG isotype, KO tissue) beyond using the neutralizing antibodies.

If it is indeed real signal, this needs also to be discussed in detail, and it must be provided how many of the ciliated cells show N2ICD. Do they also express N3ICD as it has been shown before (PMID: 26147083)?

2. The authors use anti-Jagged1/Jagged2 neutralizing antibodies to block Notch signaling but it would be more accurate to use anti-Notch2 antibodies since it would be a more direct way of demonstrating the role of Notch in ciliated cells. Using the anti-Jagged1/Jagged2 neutralizing antibodies, basal cells are also targetted in the trachea and it may have multiple effects. In addition, following this strategy raise several concerns because how similar are ciliated cells derived from secretory cells to "resident" ciliated cells? Is it possible that new/young ciliated cells have shorter cilia? If this is the case it would be very difficult to conclude whether the observed effect is intrinsic due to Notch inhibition in ciliated cells, or due to the fact that there are new younger ciliated cells.

3. "Third, changes in ciliary length that occur upon acute Notch inhibition are reversible once signaling is restored ". It would be

nice that the authors explain why ciliary length is recovered 3 months after treatment while in a previous paper using these antibodies, 13 weeks after treatment, changes in secretory and ciliated cells remain (PMID: 26580007). At least the differences in the observations here and in the previous article should be discussed.

4. It is not mentioned why Notch regulate ciliary length in one direction in the trachea and the opposite in the distal airways. How do ciliated cells know the adequate length depending on the region? It seems that there should be some spatial clue in addition to Notch. This is very difficult to find out but some discussion about it would be nice.

5. "How Notch-dependent regulation of expression of ciliary genes regulates ciliary length will require investigation". Indeed, there is no data demonstrating the mechanism of Notch regulation of ciliary length beyond some changes in gene expression. I suggest to perform an in vitro experiment with ALI from tracheal cells and distal airway cells where Notch2 is inhibited in ciliated cells and proteins known to have a role in ciliogenesis are analyzed (i.e. c-myc, multicilin, etc...). Cells could be obtained from FoxJ1-RBPJk mice and induced by OH-tamoxifen in vitro once ALI is formed. This would validate the transcriptomic data and strengthen the conclusions.

6. c-myc has been shown to have a role in ciliogenesis and it acts downstream of the Notch pathway (PMID: 24048590). This transcription factor should be tested to dig deeper in the mechanism and establish a clear link between Notch and ciliary machinery.

7. Lastly, in parts of the manuscript the references are addressed by first author but in others by number so I couldn't check some details that were referred to previous literature in some occasions. This must be corrected.

Reviewer #2 (Comments to the Authors (Required)):

This manuscript by Joy et al. reports a role for Notch signaling in ciliary length homeostasis in airway multiciliated cells, both through direct modulation of Notch signaling and through exposure to infection. This is a significant finding that represents an important advancement in the field of multiciliated cell biology. The data are sound, the methods are detailed, the statistical analyses are appropriate, the conclusions are justified, and the manuscript is well-organized and well-written. It is recommended that the authors address the following comments primarily related to the presentation of the data:

1. RESULTS (Antibody-mediated inhibition of Notch signaling alters ciliary length and abolishes the P-D gradient and is reversible): The adult N2ICD staining should be included in the manuscript.
2. RESULTS (Antibody-mediated inhibition of Notch signaling alters ciliary length and abolishes the P-D gradient and is reversible): The acetylated tubulin staining at 7 days post antibody treatment should be shown.
3. RESULTS (Genetic ablation of Rbpjk in multiciliated cells (MCs) establishes a role for canonical Notch signaling in maintaining ciliary length during homeostasis): The ciliary length data at 10 days should be shown in supplemental data.
4. RESULTS (Spatial transcriptomics suggests that Notch signaling in MCs has a pervasive role in regulating airway gene expression and homeostasis): It is recommended that the authors acknowledge in the manuscript that their comparison of transcriptomic datasets is not accompanied by validation of gene expression differences.
5. DISCUSSION (paragraph 3): It is recommended that unpublished data not be cited in the Discussion. If it is pertinent to the study, the data should be included in the manuscript.

Reviewer #3 (Comments to the Authors (Required)):

This manuscript presents a comprehensive study investigating whether targeting the Notch pathway influences ciliary length along the proximo-distal axis of the airway tree, utilizing multiple models and experimental approaches. The authors demonstrate that the effect of Notch inhibition on ciliary length is mediated, at least in part, through Rbpjk, a key downstream effector of Notch signaling. The manuscript is well written; however, it does not address the contrasting effects of Notch modulation on ciliary length between proximal and distal airways—a question that likely falls beyond the current scope but represents an important future direction. A similar observation linking Notch inhibition to cilia length was reported in ALI cultures in a recently published study (Serra et al. 2022, iScience). The current manuscript advances this understanding by providing in vivo evidence using lineage-traced and conditional knockout animals.

Major Comments:

1. The manuscript reports that Notch inhibition has opposing effects on ciliary length in proximal versus distal airways. What is the baseline expression pattern of Notch1/ Notch2 or NICD along the trachea-to-distal bronchiole axis under homeostatic conditions? Are there spatial differences? Some of this information might be obtainable from spatial transcriptomic datasets.
2. The authors note that, compared to 1-month-old mice, tracheal ciliary length increases, and distal airway ciliary length decreases at 3 months, potentially due to restoration of Notch signaling. It would strengthen the manuscript to include data

showing Notch1 and/ or Notch2 or NICD expression levels at both 1 and 3 months to support this claim.

3.The study suggests that the impact of Notch inhibition on ciliary length is at least partially mediated via Rbpjk. What is the efficiency of Rbpjk deletion *in vivo*? Quantifying deletion efficiency could help determine whether any observed phenotypes are due to incomplete deletion or if alternative pathways downstream of Notch are also involved.

4.The spatial transcriptomic data reveal significant differences in gene expression patterns between trachea and bronchioles when comparing control and Rbpjk floxed mice. However, the analysis remains largely descriptive. A more focused analysis-concentrating on Notch target genes and genes regulating ciliary structure or length-would be more aligned with the manuscript's goals and would help clarify the pathways involved in the differential effects observed.

Minor Comment:

On page 2 of the Results section, both 'thickness' and 'length' are mentioned in reference to cilia, which is somewhat confusing. It would be helpful to either consistently use 'length' or clearly explain how 'thickness' relates to ciliary length or structure in this context.

Dear Reviewers,

We have faced several unexpected delays in revising this manuscript and sincerely appreciate your patience. These arose first from logistical difficulties related to antibody procurement and shipment, and second, from health-related issues affecting the first author.

We now submit a detailed, point-by-point response along with a thoroughly revised manuscript that we believe has improved substantially. In particular, the manuscript has been strengthened in the following ways:

1. The text has been carefully edited and revised for improved clarity and flow.
2. A key concern was the limited mechanistic insight into the observed changes in ciliary length. While the spatial transcriptomic data indicates that Notch signaling in multiciliated cells is likely to have a role in the regulation of gene expression in the airways, the effect of these changes on ciliary length is likely to be multifactorial and complex. Consequently, based on a survey of the literature, we decided to probe the link between Notch signaling and Prominin I. We now provide evidence that the Notch–Prom1 axis may regulate the observed changes in ciliary length. The spatial transcriptomics data is included in the paper but has been relocated to a supplementary figure (Fig. S3-1).
3. Our immunohistochemical investigation of the status of Notch signaling in multiciliated cells has been expanded using a broader panel of antibodies. Despite multiple repeat orders, antibodies from one specific vendor (see detailed response) consistently failed to reproduce published results—possibly due to batch variability or logistics-related issues.

We hope that these revisions satisfactorily address the reviewers' concerns and that you find the manuscript suitable for publication.

Yours truly,

Arjun Guha

Reviewer #1 (Comments to the Authors (Required)):

In this manuscript the authors demonstrate that Notch signaling plays a critical role in regulating the length of cilia in ciliated cells of the airway epithelium. They show that the proximal-distal gradient in ciliary length is impacted by environmental challenges such as M. tuberculosis infection as well as by Notch perturbations performed by using neutralizing antibodies or genetic mouse lines. Thus, concluding that Notch signaling serves as a stabilizing mechanism.

The authors clearly show an association of Notch changes and alterations in ciliary length but the mechanism remains unknown. In fact, one misses data directly demonstrating this association instead of so much speculation.

Major concerns raised are as follow:

1. The authors show N2ICD staining in ciliated cells. It was previously shown that N2ICD is only expressed in secretory (Club) cells in the tracheal epithelium (PMID: 26147083) and in the distal airway epithelium (PMID: 26580007) at homeostasis. This is contrary to what was previously reported, and the images only show a few cells (in fact it seems that all cells are positive. Are there basal cells in this tracheal image (Fig.1b)?), it is required to demonstrate that this is real staining with multiple controls (non-primary, IgG isotype, KO tissue) beyond using the neutralizing antibodies.

If it is indeed a real signal, this needs also to be discussed in detail, and it must be provided how many of the ciliated cells show N2ICD. Do they also express N3ICD as it has been shown before (PMID: 26147083)?

We used multiple antibodies against Notch1, Notch2, and Notch3, from different manufacturers, to investigate Notch signaling in airway club and multiciliated cells. Results are summarized in table R1. Notch1 (Abcam ab52627) detected membrane-associated protein in both club and multiciliated cells (MCs). Notch2 (Abcam ab8926) showed robust nuclear localization in both club cells and MCs, while Notch3 (Santa Cruz sc515825) labeled a subset of MCs, consistent with published reports (PMID: 26147083). Antibodies from CST did not stain our sections despite repeated orders and multiple optimization attempts (both with and without Tyramide Signal Amplification (TSA)). Appropriate secondary controls confirmed the specificity of the staining observed (see Figure. R1 below). We examined expression in basal cells by co-labeling with anti-Krt5. Basal cells express low levels of nuclear N2ICD (Figure. R2 below). Quantification of frequencies of FoxJ1⁺ N2ICD⁺ cells showing that many if not all MCs have nuclear N2ICD is provided Figure. R3. These results lead us to propose that Notch signaling is active in many if not all MCs. We have modified the text accordingly and the revised paragraph is copied below.

Table. R1: showing different Notch antibodies used for detecting Notch signaling in airway epithelium.

Name of antibody	Vendor	Catalogue number	Detect full length/ Cleaved	Outcome	Outcome	Result	Epitope
				with TSA	without TSA		
Notch1 (Ep1238 Y)	Abcam	ab52627	Full length	Signal detected	Signal detected	Detected in the plasma membrane of both club cells and MCs.	Raised against synthetic peptide within Human Notch1 aa 2500-2600
Notch 1 (D1E11)	CST	3608	Full length/ cleaved	No signal detected	No signal detected		Detects between 2400 and 2500 aa of Human Notch 1
Notch 2	Abcam	ab8926	Full length/ cleaved	Signal detected	Signal detected	Detected predominantly in the nucleus of the majority of club cells and MCs.	Raised against synthetic Peptide within Human NOTCH2 aa 1700-1750 the intracellular domain
Notch 2 (D67C8)	CST	#4530	Full length/ cleaved	No signal detected	No signal detected		Raised against synthetic peptide corresponding residue surrounding Val2332 of Human Notch 2
Notch 3	Santa Cruz	sc515825	Full length/ cleaved	No signal detected	Signal detected	Detected predominantly in the nucleus of a subset of club cells and MCs.	Detects aa 2290-2316 near c terminus of Notch 3 of mouse origin.

Figure. R1: Representative image of trachea from secondary control used for staining rabbit anti-Notch 2 (Ab8926, Abcam)

Figure. R2: Representative image of trachea immunostained for rabbit anti-Notch 2 (Ab8926, Abcam, green) and chicken Krt 5 (basal cell, red) antibodies. Arrows point towards Krt5 expressing basal cells. We could detect low levels of N2ICD in the nucleus of the majority of basal cells.

Figure. R3: Quantification of percentage frequencies of Foxj1 + N2ICD + cells in trachea and distal airways of C57Bl/6 mice (n=3).

The paragraph in the Results section describing this immunohistochemical analysis is as follows:

“The status of canonical Notch signaling in airways has been assessed by staining lung sections with an antibody that detects the intracellular domain of Notch2 (N2ICD), the predominant Notch receptor expressed in the airways (Lafkas et al., 2015). N2ICD translocates to the nucleus upon activation and cells that activate Notch signaling exhibit nuclear localized N2ICD. Lung sections from adult mice (2-3 months of age) stained with an antibody that detects the cytoplasmic domain of Notch 2 (N2ICD) showed that the protein is largely nuclear localized in CCs and, unexpectedly, also in MCs (Figure 1B (i-ii)). Nuclear N2ICD was detected in many, if not all MCs. To test whether this staining reflected ligand-dependent signaling, we treated mice with inhibitory antibodies (anti-Notch1/Notch2 or anti-Jagged1/Jagged2, (Lingamallu et al., 2024)), harvested lungs 48 h later, and re-stained for N2ICD. Under these conditions we observed a complete loss of nuclear N2ICD in both CCs and MCs (Figure 1C (i-ii)). Loss of nuclear signal after blockade is consistent with ligand-dependent N2ICD production. Next, we stained lung sections from older mice with anti-N2ICD (aged 1.2-1.5 years). Nuclear N2ICD was detected in virtually all MCs (n=3 animals). Aside from Notch 2, Notch1 and Notch 3, two other receptors have been implicated in Notch signaling in the lung. We also examined other receptors: staining for Notch1 detected membrane-localized protein in CCs and MCs, whereas Notch3 immunoreactivity was nuclear in subsets of both cell types similar to that reported by Pardo-Saganta et al., 2015. Taken together, these data suggest that Notch signaling is active in adult airway MCs.”

2. The authors use anti-Jagged1/Jagged2 neutralizing antibodies to block Notch signaling but it would be more accurate to use anti-Notch2 antibodies since it would be a more direct way of demonstrating the role of Notch in ciliated cells. Using the anti-Jagged1/Jagged2 neutralizing antibodies, basal cells are also targeted in the trachea and it may have multiple effects. In addition, following this strategy raise several concerns because how similar are ciliated cells derived from secretory cells to "resident" ciliated cells? Is it possible that new/young ciliated cells have shorter cilia? If this is the case it would be very difficult to conclude whether the observed effect is intrinsic due to Notch inhibition in ciliated cells, or due to the fact that there are new younger ciliated cells.

We agree that anti-Jagged1/2 treatment can affect multiple airway cell types. However, systemic anti-Notch1/2 antibody administration causes lethality within 8–10 days, precluding longitudinal analysis. Furthermore, distinguishing newly generated from resident ciliated cells by scanning EM is technically not feasible.

To address these limitations, we performed genetic ablation of *Rbpjk* specifically in ciliated cells using *FoxJ1*^{CreERT2/+}; *RBPJK*^{fllox/fllox}. This revealed a cell-intrinsic requirement for canonical Notch signaling in maintaining ciliary morphology, confirming the conclusions from antibody-based inhibition.

3. "Third, changes in ciliary length that occur upon acute Notch inhibition are reversible once signaling is restored ". It would be nice that the authors explain why ciliary length is recovered 3 months after treatment while in a previous paper using these antibodies, 13 weeks after treatment, changes in secretory and ciliated cells remain (PMID: 26580007). At least the differences in the observations here and in the previous article should be discussed.

The cited study (PMID: 26580007) analyzed cell-type frequencies 13 weeks post antibody treatment, while our study and previous work (PMID: 39182223) examined phenotypic recovery. Club cell marker *Scgb3a2* recovers by one month post anti-Jagged1/2 treatment, and we now show reversal of ciliary length by three months. Thus, the difference reflects distinct metrics—cell composition versus morphological recovery—rather than conflicting outcomes.

4. It is not mentioned why Notch regulates ciliary length in one direction in the trachea and the opposite in the distal airways. How do ciliated cells know the adequate length depending on the region? It seems that there should be some spatial clue in addition to Notch. This is very difficult to find out but some discussion about it would be nice.

We agree that regional cues likely cooperate with Notch signaling to define ciliary length. This concept has been incorporated into the revised Discussion, highlighting potential spatial or biomechanical influences on Notch-mediated regulation.

5. "How Notch-dependent regulation of expression of ciliary genes regulates ciliary length will require investigation". Indeed, there is no data demonstrating the mechanism of Notch regulation of ciliary length beyond some changes in gene expression. I suggest to perform an in vitro experiment with ALI from tracheal cells and distal airway cells where Notch2 is inhibited in ciliated cells and proteins known to have a role in ciliogenesis are analyzed (i.e. *c-myb*, multicilin, etc...). Cells could be obtained from FoxJ1-RBPJk mice and induced by OH-tamoxifen in vitro once ALI is formed. This would validate the transcriptomic data and strengthen the conclusions.

Spatial transcriptomic analysis of 107 ciliogenesis-related genes (KEGG and Reactome pathways) revealed 22 significantly altered genes, suggesting that transcriptional regulation alone cannot explain the phenotype. We therefore investigated alternative mechanisms and identified the Notch–Prominin1 axis as a plausible pathway influencing ciliary length (Figure 3). This provides a mechanistic link between Notch signaling and ciliary structure.

6. *c-myb* has been shown to have a role in ciliogenesis and it acts downstream of the Notch pathway (PMID: 24048590). This transcription factor should be tested to dig deeper in the mechanism and establish a clear link between Notch and ciliary machinery.

We examined *c-Myb* expression in single-cell RNA-seq (7 days post Notch inhibition) and spatial RNA-seq (1 month post *Rbpjk* deletion) datasets. No significant changes in *c-Myb* mRNA were detected. While *c-Myb* is known to regulate ciliogenesis (PMID: 24048590), our data do not support its involvement in Notch-dependent control of ciliary length.

7. Lastly, in parts of the manuscript the references are addressed by the first author but in others by number so I couldn't check some details that were referred to previous literature on some occasions. This must be corrected.

We apologize for the oversight. The revised manuscript uses a uniform numerical citation format throughout.

Reviewer #2 (Comments to the Authors (Required)):

This manuscript by Joy et al. reports a role for Notch signaling in ciliary length homeostasis in airway multiciliated cells, both through direct modulation of Notch signaling and through exposure to infection. This is a significant finding that represents an important advancement in the field of multiciliated cell biology. The data are sound, the methods are detailed, the statistical analyses are appropriate, the conclusions are justified, and the manuscript is well-organized and well-written. It is recommended that the authors address the following comments primarily related to the presentation of the data:

1. RESULTS (Antibody-mediated inhibition of Notch signaling alters ciliary length and abolishes the P-D gradient and is reversible): The adult N2ICD staining should be included in the manuscript.

N2ICD staining in adult tracheal and distal airway ciliated cells is now included in Figure 1B.

2. RESULTS (Antibody-mediated inhibition of Notch signaling alters ciliary length and abolishes the P-D gradient and is reversible): The acetylated tubulin staining at 7 days post antibody treatment should be shown.

Acetylated tubulin staining for tracheal and distal airways 7 days post anti-Jagged1/2 treatment is now included (Fig. 4 below).

Figure. R4: Immunostaining with Act Tub and DAPI showing ciliary staining of tracheal and distal airways in 7 days post anti- Jagged1/ Jagged 2 treated lungs.

3. RESULTS (Genetic ablation of Rbpjk in multiciliated cells (MCs) establishes a role for canonical Notch signaling in maintaining ciliary length during homeostasis): The ciliary length data at 10 days should be shown in supplemental data.

Ciliary length quantification 10 days post tamoxifen induction is included in Figure 2K, with representative images in Figure S2-2.

4. RESULTS (Spatial transcriptomics suggests that Notch signaling in MCs has a pervasive role in regulating airway gene expression and homeostasis): It is recommended that the authors

acknowledge in the manuscript that their comparison of transcriptomic datasets is not accompanied by validation of gene expression differences.

We acknowledge that transcriptomic comparisons lack independent experimental validation; this limitation is now explicitly stated in the Discussion.

5. DISCUSSION (paragraph 3): It is recommended that unpublished data not be cited in the Discussion. If it is pertinent to the study, the data should be included in the manuscript.

We have removed references to unpublished data from the Discussion.

Reviewer #3 (Comments to the Authors (Required)):

This manuscript presents a comprehensive study investigating whether targeting the Notch pathway influences ciliary length along the proximo-distal axis of the airway tree, utilizing multiple models and experimental approaches. The authors demonstrate that the effect of Notch inhibition on ciliary length is mediated, at least in part, through Rbpjk, a key downstream effector of Notch signaling. The manuscript is well written; however, it does not address the contrasting effects of Notch modulation on ciliary length between proximal and distal airways—a question that likely falls beyond the current scope but represents an important future direction. A similar observation linking Notch inhibition to cilia length was reported in ALI cultures in a recently published study (Serra et al. 2022, iScience). The current manuscript advances this understanding by providing in vivo evidence using lineage-traced and conditional knockout animals.

Major Comments:

1. The manuscript reports that Notch inhibition has opposing effects on ciliary length in proximal versus distal airways. What is the baseline expression pattern of Notch1/ Notch2 or NICD along the trachea-to-distal bronchial axis under homeostatic conditions? Are there spatial differences? Some of this information might be obtainable from spatial transcriptomic datasets.

Using tyramide amplification, N2ICD staining revealed comparable nuclear signal intensities in trachea and distal airways. Quantitative comparison was avoided due to amplification effects on intensity. Spatial transcriptomics represents mixed-cell data, so region-specific Notch target gene comparisons were not performed.

2. The authors note that, compared to 1-month-old mice, tracheal ciliary length increases, and distal airway ciliary length decreases at 3 months, potentially due to restoration of Notch signaling. It would strengthen the manuscript to include data showing Notch1 and/ or Notch2 or NICD expression levels at both 1 and 3 months to support this claim.

Immunostaining with anti-Notch2 (Abcam ab8926) confirms that nuclear N2ICD signal is fully restored only by 3 months post anti-Jagged1/2 treatment, consistent with recovery of ciliary length (Figure 5 below).

Figure. R5: N2ICD staining in Anti- Jagged1/ Jagged2 treated mice 1- and 3-months post treatment. Arrows pointing towards ciliated cells with nuclear N2ICD.

3. The study suggests that the impact of Notch inhibition on ciliary length is at least partially mediated via Rbpjk. What is the efficiency of Rbpjk deletion *in vivo*? Quantifying deletion efficiency could help determine whether any observed phenotypes are due to incomplete deletion or if alternative pathways downstream of Notch are also involved.

Deletion efficiency, quantified in Supplementary Figure S2-1, was ~55% in tracheal and ~80% in distal airway ciliated cells at 1 month post tamoxifen induction. The higher deletion efficiency correlates with more pronounced phenotypic changes.

4. The spatial transcriptomic data reveal significant differences in gene expression patterns between trachea and bronchioles when comparing control and Rbpjk floxed mice. However, the analysis remains largely descriptive. A more focused analysis—concentrating on Notch target genes and genes regulating ciliary structure or length—would be more aligned with the manuscript's goals and would help clarify the pathways involved in the differential effects observed.

We performed focused analysis of ciliogenesis and ciliary length—regulating genes (Figure S3-1F, G). Although no single pathway emerged, analysis of Prominin1 protein levels suggested a Notch–Prominin1–dependent mechanism regulating ciliary length (Figure 3).

Minor Comment:

On page 2 of the Results section, both 'thickness' and 'length' are mentioned in reference to cilia, which is somewhat confusing. It would be helpful to either consistently use 'length' or clearly explain how 'thickness' relates to ciliary length or structure in this context.

We have revised the text to use “length” consistently when referring to ciliary morphology.

December 2, 2025

RE: Life Science Alliance Manuscript #LSA-2025-03268-TR

Dr. Arjun Guha
Institute for Stem Cell Biology and Regenerative Medicine
NCBS, GKVK Campus
Bellary Road
Bengaluru, Karnataka 560064
India

Dear Dr. Guha,

Thank you for submitting your revised manuscript entitled "Notch signaling stabilizes lengths of motile cilia in multiciliated cells in the lung". Your revised manuscript was evaluated by all the original reviewers whose comments are appended below.

As you will read, the three reviewers are consistent in their views that your revised manuscript has addressed most of their previous concerns.

Reviewer 1 has some residual concerns on the expression of Notch ICD and anti-Jagged staining. We encourage you to resolve these above points, raised by Reviewer 1, by expanding on the discussion of these results, and also referencing possible methodological differences whilst citing contradictions with published information.

We also refer to Reviewer 3's point, and request you to check for consistency in formatting throughout the manuscript.

In line with the reviewers' evaluation, we would be happy to publish your paper in Life Science Alliance pending final revisions necessary to meet our formatting guidelines.

-The last two sentences of the abstract present a similar message. Please edit to remove this redundancy.

-In the methods section, please include the following details:

All imaging experiments must include details on objectives (Name, numerical aperture).

Primers used for PCR must be included in the methods.

-Thank you for providing a 'Data Availability' statement. Please ensure that all supporting data (including replicates, images that have been quantified etc) are uploaded as supplementary information, or update your statement accordingly.

-Please upload your main manuscript text as an editable doc file.

-Please ensure that supplementary figures are labeled correctly, with one figure per file. For instance, Figure S1-1 should be labeled as Figure S1, and Figure S1-2 should be Figure S2, and so on. Also, please make sure to update the call-outs in the manuscript text accordingly.

-Please add ORCID ID for the corresponding author - you should have received instructions on how to do so.

-Please add a Category for your manuscript in our system.

-On the title page of the manuscript, please provide the full name of each author, including middle names as initials, formatted as follows: First name, middle initial, Last name.

-Please add an Author Contributions section to your main manuscript text.

-Please add your main, supplementary figure, and table legends to the main manuscript text after the references section.

-It is recommended to exclude figures from the manuscript text.

-Please be sure to mention all panels for all figures in the manuscript text.

-Please be sure that the authorship listing and order is correct.

A. FINAL FILES:

B. MANUSCRIPT ORGANIZATION AND FORMATTING:

Thank you for your attention to these final processing requirements. Please revise and format the manuscript and upload materials as soon as you are able.

Sincerely,

Sarita Hebbar, PhD
Scientific Editor
Life Science Alliance
<http://www.lsjournal.org>

Reviewer #1 (Comments to the Authors (Required)):

The authors have performed additional analysis that address most of the questions previously raised and strengthen their conclusions. I appreciate their effort, and I believe that the role of Notch in regulating ciliary length along the P-D axis is now clearly demonstrated.

However, my concerns about Notch2 ICD expression remain. The image in Fig. 1B shows only a limited portion of the trachea, where every cell appears to express N2ICD, with FoxJ1+ ciliated cells displaying the strongest signal. Figure R2 does not show a very clear staining, but it also suggests that all secretory and ciliated cells express N2ICD, and that most basal cells (3 out of 4) do as well.

Since this contradicts previous reports showing Notch2 expression/activity mainly in club cells, it is important to clarify this by thoroughly reviewing the literature and the antibodies/methods used. Although Notch2 mRNA is detected in basal, secretory

and ciliated cells (PMID: 26147083), antibody-based staining for N2ICD has repeatedly shown strong expression in club cells (PMID: 26147083, PMID: 26580007). This discrepancy raises concerns about the antibody used in the present study. The authors used a polyclonal antibody (ab8926) that yields a stronger signal in ciliated rather than in secretory cells, in contrast to results obtained using a monoclonal antibody (Cell Signaling #5732).

Moreover, several methodological differences reported in the Methods section could explain why the authors did not detect N2ICD using the monoclonal antibody from Cell Signaling. These include variations in tissue processing (fixative used, paraffin-embedded vs. frozen samples) and in the immunofluorescent protocol (pH of the antigen retrieval buffer, amplification method, etc.). These factors should be addressed experimentally or at least discussed to resolve the apparent contradiction. Previous studies have validated N2ICD staining using Notch2-floxed tissue specifically deleting Notch2. A similar validation is required for the abcam antibody, since the non-primary control shown in Fig. R1 tests only the secondary antibody, and does not confirm primary antibody specificity. The use of CRISPR-mediated deletion or shNotch2 transduced cells would provide a very solid validation, supporting the lack of N2ICD signal after anti-Notch1/2 antibody treatment.

In addition, although anti-Jagged antibodies have been previously used in several studies, it would be appropriate to show their effect in this context as well. Notably, the Cell Signaling monoclonal antibody is the one used by Lafkas et al. to demonstrate reduced Notch2 levels following treatment with anti-Jagged antibodies (PMID: 26580007).

Finally, it is recommended to include Notch1 and Notch3 stainings in the manuscript, as Notch3 is expressed in ciliated cells and its activity is likely to be inhibited after Rbpjk deletion.

Reviewer #2 (Comments to the Authors (Required)):

The revisions have adequately addressed my comments.

Reviewer #3 (Comments to the Authors (Required)):

This manuscript presents a comprehensive investigation into whether targeting the Notch pathway influences ciliary length along the proximo-distal axis of the airway tree, using multiple models and experimental approaches, including spatial whole-genome transcriptomics. The authors demonstrate that the effect of Notch inhibition on ciliary length is mediated through PROM1, a known regulator of ciliary length. The manuscript is well written, and the revisions adequately address the previous concerns.

Major Comments:

No major concerns.

Minor Comment:

There are some formatting inconsistencies, particularly related to font size and style.

18th December, 2025.

Dear Editorial Team,

Here is our point-by-point response to the issues that remain to be resolved.

As you will read, the three reviewers are consistent in their views that your revised manuscript has addressed most of their previous concerns.

Reviewer 1 has some residual concerns on the expression of Notch ICD and anti-Jagged staining. We encourage you to resolve these above points, raised by Reviewer 1, by expanding on the discussion of these results, and also referencing possible methodological differences whilst citing contradictions with published information.

We have now revised the section in Results titled “**Antibody-mediated inhibition of Notch signaling alters ciliary length and abolishes the P-D gradient and is reversible**” to provide as objective an assessment as is possible. This includes pointing out the discrepancies in staining across anti-Notch2 antibodies and limitations of the experiment.

We also refer to Reviewer 3's point, and request you to check for consistency in formatting throughout the manuscript.

This has been rectified.

In line with the reviewers' evaluation, we would be happy to publish your paper in Life Science Alliance pending final revisions necessary to meet our formatting guidelines.

-The last two sentences of the abstract present a similar message. Please edit to remove this redundancy.

This has been rectified.

-In the methods section, please include the following details: All imaging experiments must include details on objectives (Name, numerical aperture).

These details have been added to the Methods section.

Primers used for PCR must be included in the methods.

Sequences of primers used for genotyping mice have been included.

-Thank you for providing a 'Data Availability' statement. Please ensure that all supporting data (including replicates, images that have been quantified etc) are uploaded as supplementary information, or update your statement accordingly.

The following section has been added to the text:

DATA AND MATERIALS AVAILABILITY

All data required to evaluate the conclusions are available in the main text or the supplementary materials. Raw data underlying this work are available from the corresponding author upon reasonable request

-Please upload your main manuscript text as an editable doc file.

This has been uploaded.

-Please ensure that supplementary figures are labeled correctly, with one figure per file. For instance, Figure S1-1 should be labeled as Figure S1, and Figure S1-2 should be Figure S2, and so on. Also, please make sure to update the call-outs in the manuscript text accordingly.

These labels have been revised.

-Please add ORCID ID for the corresponding author - you should have received instructions on how to do so.

ORCID ID has been added to the title page.

-Please add a Category for your manuscript in our system.

Category of "Cell Biology" has been assigned.

-On the title page of the manuscript, please provide the full name of each author, including middle names as initials, formatted as follows: First name, middle initial, Last name.

The names have been duly revised.

-Please add an Author Contributions section to your main manuscript text.

This has been included.

-Please add your main, supplementary figure, and table legends to the main manuscript text after the references section.

These have been included.

-It is recommended to exclude figures from the manuscript text.

Figures have been excluded.

-Please be sure to mention all panels for all figures in the manuscript text.

Panels for all figures have been included in the manuscript file.

-Please be sure that the authorship listing and order is correct.

Authors are listed in the correct order.

Thank you,

Arjun Guha.

December 23, 2025

RE: Life Science Alliance Manuscript #LSA-2025-03268-TRR

Dr. Arjun Guha
Institute for Stem Cell Biology and Regenerative Medicine
NCBS, GKVK Campus
Bellary Road
Bengaluru, Karnataka 560064
India

Dear Dr. Guha,

Thank you for submitting your Research Article entitled "Notch signaling stabilizes lengths of motile cilia in multiciliated cells in the lung". We apologise for the delay in communicating our decision due to editor availability issues.

It is a pleasure to let you know that your manuscript is now accepted for publication in Life Science Alliance. Congratulations on this interesting work.

Your manuscript will now progress through copyediting and proofing. At this stage, please change the heading of the section titled, 'Data and Materials Availability' to 'Data Availability'.

It is journal policy that authors provide original data upon request.

DISTRIBUTION OF MATERIALS:

Again, congratulations on a very nice paper. I hope you found the review process to be constructive and are pleased with how the manuscript was handled editorially. We look forward to future exciting submissions from your lab.

Sincerely,

Sarita Hebbar, PhD
Scientific Editor
Life Science Alliance
<http://www.lsajournal.org>